# Genetic insights into ossification of the posterior longitudinal ligament of the spine

Yoshinao Koike[1,2,3†], Masahiko Takahata[3*†], Masahiro Nakajima[1†], Nao Otomo[1,2,4], Hiroyuki Suetsugu[1,2,5], Xiaoxi Liu[2], Tsutomu Endo[3], Shiro Imagama[6], Kazuyoshi Kobayashi[6], Takashi Kaito[7], Satoshi Kato[8], Yoshiharu Kawaguchi[9], Masahiro Kanayama[10], Hiroaki Sakai[11], Takashi Tsuji[4,12], Takeshi Miyamoto[4,13], Hiroyuki Inose[14], Toshitaka Yoshii[14], Masafumi Kashii[7], Hiroaki Nakashima[6], Kei Ando[6], Yuki Taniguchi[15], Kazuhiro Takeuchi[16], Shuji Ito[2,17], Kohei Tomizuka[2], Keiko Hikino[18], Yusuke Iwasaki[19], Yoichiro Kamatani[20], Shingo Maeda[21], Hideaki Nakajima[22], Kanji Mori[23], Atsushi Seichi[24], Shunsuke Fujibayashi[25], Tsukasa Kanchiku[26], Kei Watanabe[27], Toshihiro Tanaka[28], Kazunobu Kida[29], Sho Kobayashi[30], Masahito Takahashi[31], Kei Yamada[32], Hiroshi Takuwa[1,17], Hsing-Fang Lu[1,33], Shumpei Niida[34], Kouichi Ozaki[35], Yukihide Momozawa[19], Genetic Study Group of Investigation Committee on Ossification of the Spinal Ligaments, Masashi Yamazaki[36], Atsushi Okawa[14], Morio Matsumoto[4], Norimasa Iwasaki[3], Chikashi Terao[2*], Shiro Ikegawa[1,3*]

*For correspondence:
takamasa@med.hokudai.ac.jp
(MT);
chikashi.terao@riken.jp (CT);
sikegawa@ims.u-tokyo.ac.jp (SI)

†These authors contributed
equally to this work

Group author details:
Genetic Study Group of
Investigation Committee on
Ossification of the Spinal
Ligaments See page 22

Competing interest: See page
23

Reviewing Editor: Cheryl
Ackert-Bicknell, University of
Colorado, United States

[1]Laboratory for Bone and Joint Diseases, Center for Integrative Medical Sciences, RIKEN, Tokyo, Japan; [2]Laboratory for Statistical and Translational Genetics, Center for Integrative Medical Sciences, RIKEN, Yokohama, Japan; [3]Department of Orthopedic Surgery, Hokkaido University Graduate School of Medicine, Sapporo, Japan; [4]Department of Orthopedic Surgery, Keio University School of Medicine, Nagoya, Japan; [5]Department of Orthopaedic Surgery, Graduate School of Medical Sciences, Kyushu University, Fukuoka, Japan; [6]Department of Orthopedics, Nagoya University Graduate School of Medicine, Nagoya, Japan; [7]Department of Orthopaedic Surgery, Osaka University Graduate School of Medicine, Suita, Japan; [8]Department of Orthopaedic Surgery, Graduate School of Medical Science, Kanazawa University, Kanazawa, Japan; [9]Department of Orthopaedic Surgery, Toyama University, Toyama, Japan; [10]Department of Orthopedics, Hakodate Central General Hospital, Hakodate, Japan; [11]Department of Orthopaedic Surgery, Spinal Injuries Center, Iizuka, Japan; [12]Department of Spine and Spinal Cord Surgery, Fujita Health University, Toyoake, Japan; [13]Department of Orthopedic Surgery, Kumamoto University, Kumamoto, Japan; [14]Department of Orthopaedic Surgery, Tokyo Medical and Dental University, Tokyo, Japan; [15]Department of Orthopaedic Surgery, Faculty of Medicine, The University of Tokyo, Tokyo, Japan; [16]Department of Orthopaedic Surgery, National Okayama Medical Center, Okayama, Japan; [17]Department of Orthopedic Surgery, Shimane University Faculty of Medicine, Izumo, Japan; [18]Laboratory for Pharmacogenomics, Center for Integrative Medical Sciences, RIKEN, Yokohama, Japan; [19]Laboratory for Genotyping Development, Center for Integrative Medical Sciences, RIKEN, Yokohama, Japan; [20]Laboratory for Statistical Analysis, Center for Integrative Medical Sciences, RIKEN, Yokohama, Japan; [21]Department of Bone and Joint Medicine, Graduate School of Medical and Dental Sciences, Kagoshima University, Kagoshima, Japan; [22]Department of Orthopaedics

and Rehabilitation Medicine, Faculty of Medical Sciences, University of Fukui, Fukui, Japan; [23]Department of Orthopaedic Surgery, Shiga University of Medical Science, Otsu, Japan; [24]Department of Orthopedics, Jichi Medical University, Shimotsuke, Japan; [25]Department of Orthopaedic Surgery, Graduate School of Medicine, Kyoto University, Kyoto, Japan; [26]Department of Orthopedic Surgery, Yamaguchi University Graduate School of Medicine, Ube, Japan; [27]Department of Orthopaedic Surgery, Niigata University Medical and Dental General Hospital, Nankoku, Japan; [28]Department of Orthopaedic Surgery, Hirosaki University Graduate School of Medicine, Hirosaki, Japan; [29]Department of Orthopaedic Surgery, Kochi Medical School, Nankoku, Japan; [30]Department of Orthopaedic Surgery, Hamamatsu University School of Medicine, Hamamatsu, Japan; [31]Department of Orthopaedic Surgery, Kyorin University School of Medicine, Tokyo, Japan; [32]Department of Orthopaedic Surgery, Kurume University School of Medicine, Obu, Japan; [33]Million-Person Precision Medicine Initiative, China Medical University Hospital, Taichung, Taiwan; [34]Core Facility Administration, Research Institute, National Center for Geriatrics and Gerontology, Obu, Japan; [35]Medical Genome Center, Research Institute, National Center for Geriatrics and Gerontology, Obu, Japan; [36]Department of Orthopaedic Surgery, Faculty of Medicine, University of Tsukuba, Tsukuba, Japan

**Abstract** Ossification of the posterior longitudinal ligament of the spine (OPLL) is an intractable disease leading to severe neurological deficits. Its etiology and pathogenesis are primarily unknown. The relationship between OPLL and comorbidities, especially type 2 diabetes (T2D) and high body mass index (BMI), has been the focus of attention; however, no trait has been proven to have a causal relationship. We conducted a meta-analysis of genome-wide association studies (GWASs) using 22,016 Japanese individuals and identified 14 significant loci, 8 of which were previously unreported. We then conducted a gene-based association analysis and a transcriptome-wide Mendelian randomization approach and identified three candidate genes for each. Partitioning heritability enrichment analyses observed significant enrichment of the polygenic signals in the active enhancers of the connective/bone cell group, especially H3K27ac in chondrogenic differentiation cells, as well as the immune/hematopoietic cell group. Single-cell RNA sequencing of Achilles tendon cells from a mouse Achilles tendon ossification model confirmed the expression of genes in GWAS and post-GWAS analyses in mesenchymal and immune cells. Genetic correlations with 96 complex traits showed positive correlations with T2D and BMI and a negative correlation with cerebral aneurysm. Mendelian randomization analysis demonstrated a significant causal effect of increased BMI and high bone mineral density on OPLL. We evaluated the clinical images in detail and classified OPLL into cervical, thoracic, and the other types. GWAS subanalyses identified subtype-specific signals. A polygenic risk score for BMI demonstrated that the effect of BMI was particularly strong in thoracic OPLL. Our study provides genetic insight into the etiology and pathogenesis of OPLL and is expected to serve as a basis for future treatment development.

### Editor's evaluation

This study builds on previous work to explore the genetic causes of ossification of the posterior longitudinal ligament of the spine (OPLL). A meta-Genome wide association study is conducted to increase detection power and a disease subtype analysis is completed that provides new information on if all sites of OPLL have uniform causes. Using additional open-source data, the GWAS results are explored further to find putatively causative genes and to explore causative co-existing conditions such as obesity for OPLL. Overall this study is by far the most complete genetic exploration of this disease to date and is instructive for future studies that will lead to treatments for this condition.

## Introduction

Ossification of the posterior longitudinal ligament of the spine (OPLL) is an incurable disease with progressive heterotopic ossification. It can occur at any spine level from the cervical to the lumbar spine, and ossified ligaments compress the spinal cord and roots, leading to a severe neurological deficit (*Matsunaga and Sakou, 2012*). OPLL is a common disease; however, its frequency varies depending on the region of the world; high in Asian countries (0.4–3.0%), especially Japan (1.9–4.3%), compared with Europe and the United States (0.1–1.7%) (*Matsunaga and Sakou, 2012*; *Ohtsuka et al., 1987*; *Sohn and Chung, 2013*; *Yoshimura et al., 2014*). However, its etiology and pathogenesis remain unknown. Histological studies suggest that OPLL develops through endochondral ossification (*Sato et al., 2007*; *Sugita et al., 2013*). In recent years, OPLL has been reported to have different clinical characteristics depending on the affected region: higher body mass index (BMI), earlier-onset of symptoms, and more diffuse progression of OPLL over the entire spine in the thoracic type of OPLL (T-OPLL) than in the cervical type (C-OPLL) (*Endo et al., 2020*; *Hisada et al., 2022*). This fact suggests that there may be differences in etiology and pathogenesis for each subtype of OPLL. Currently, there is no therapeutic or preventive measure for OPLL other than surgery to decompress the spinal cord and roots. Therefore, it is necessary to clarify its etiology and pathogenesis to develop effective measures to prevent and treat OPLL.

OPLL is assumed to be a polygenic disease where complex genetic and environmental factors interact. Epidemiological studies have reported the relationship between OPLL and various other traits, especially type 2 diabetes (T2D) (*Akune et al., 2001*; *Kobashi et al., 2004*), high BMI (*Hou et al., 2017*; *Kobashi et al., 2004*), low inorganic phosphate, X-linked hypophosphatemic rickets (*Chesher et al., 2018*), and increased C-reactive protein (*Kawaguchi et al., 2017*). Of these traits, T2D has been the focus of attention for a long time (*Akune et al., 2001*; *Kobashi et al., 2004*). Furthermore, because of the high incidence within families and close relatives in previous epidemiological studies, genetic factors have long been considered in OPLL development (*Matsunaga et al., 1999*; *Sakou et al., 1991*; *Terayama, 1989*), although there are no previous papers evaluating the heritability of OPLL such as twin studies. To understand the genetic factors associated with OPLL, we previously conducted a genome-wide association study (GWAS) and found six significant loci (*Nakajima et al., 2014*). In subsequent in silico and in vitro functional studies, we identified *RSPO2* as a susceptibility gene for OPLL, and the role of Wnt signaling in the pathogenesis of OPLL was clarified (*Nakajima et al., 2016*). However, the pathogenesis of this condition remains largely unknown.

In this study, to clarify the etiology and pathogenesis of OPLL, we conducted a meta-analysis of GWASs and various post-GWAS analyses. We identified 14 significant loci, including 8 previously unreported susceptibility loci. Using a gene-based analysis (*de Leeuw et al., 2015*) and summary data-based Mendelian randomization (SMR) (*Zhu et al., 2016*), we identified three candidate genes for each. Using a genetic correlation analysis and a subsequent Mendelian randomization (MR) study, we identified a causal effect of high BMI on OPLL. A polygenic risk score (PRS) of BMI demonstrated the heterogeneity of the impact of obesity on OPLL subtypes.

## Results

### Novel susceptibility loci in OPLL

We conducted three GWASs (set 1–3) in the Japanese population (*Supplementary file 1*). After quality control of single-nucleotide polymorphism (SNP) genotyping data, we performed imputation and association analyses independently for each GWAS. Subsequently, we performed a fixed-effects meta-analysis combining the three GWASs (ALL-OPLL: a total of 2010 cases and 20,006 controls; *Figure 1—figure supplement 1*) and identified 12 genome-wide significant loci ($p<5.0 \times 10^{-8}$) (*Figure 1*). The genomic inflation factor ($\lambda$ GC) was 1.11 and showed slight inflation in GWAS; however, the intercept in linkage disequilibrium (LD) score regression (*Bulik-Sullivan et al., 2015*) was 1.03, indicating that inflation of the statistics was mainly from polygenicity and minimal biases of the association results (*Figure 1—figure supplement 2*).

Next, we conducted a stepwise conditional analysis to detect multiple independent signals. We detected two additional independent signals that showed genome-wide significance after conditioning (*Supplementary file 2*): rs35281060 (12p12.3, $p=1.04 \times 10^{-10}$) and rs1038666 (12p11.22, $p=2.37 \times 10^{-10}$) (*Figure 1—figure supplement 3F–I*). We also detected one additional signal (rs61915977,

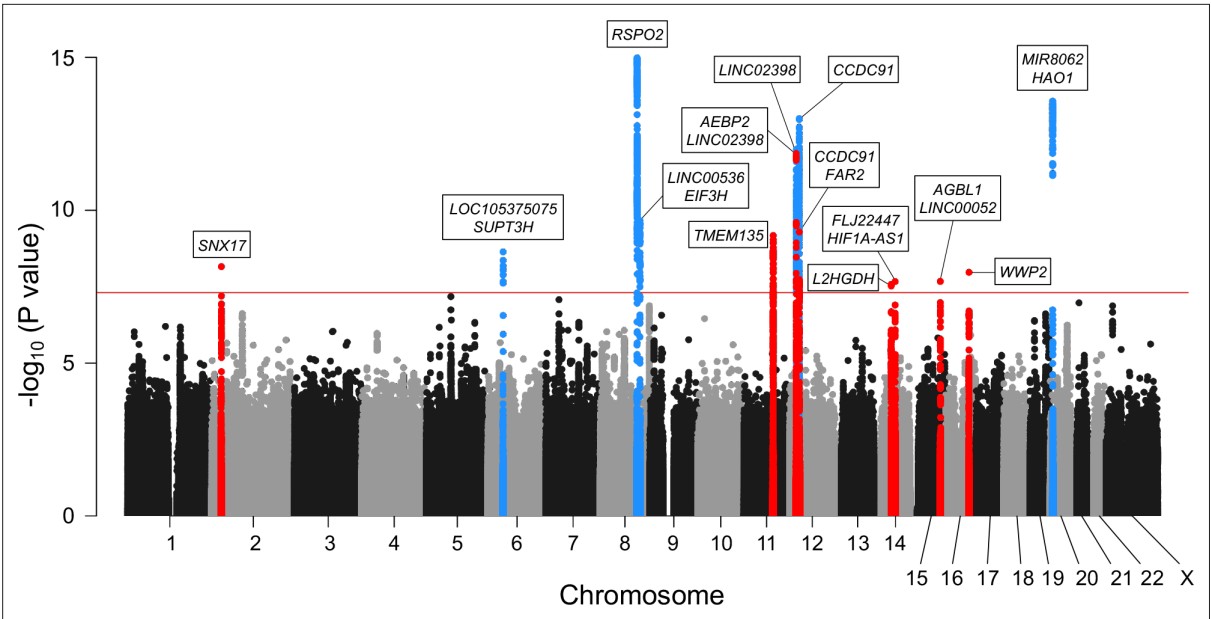

**Figure 1.** Meta-analysis of genome-wide association studies (GWAS) identified 14 significant loci in ossification of the posterior longitudinal ligament of the spine (OPLL). Manhattan plot showing the -log10 p-value for each single-nucleotide polymorphism (SNP) in the meta-analysis. The values were plotted against the respective chromosomal positions. The horizontal red line represents the genome-wide significance threshold (p=$5.0 \times 10^{-8}$). Red and blue points represent the SNPs in the new and known loci, respectively.

The online version of this article includes the following figure supplement(s) for figure 1:

**Figure supplement 1.** The overview of the genome-wide association study meta-analysis.

**Figure supplement 2.** A quantile–quantile plot of meta-analysis of genome-wide association studies.

**Figure supplement 3.** Regional association plots for 14 susceptibility loci for ossification of the posterior longitudinal ligament of the spine (OPLL).

**Figure supplement 4.** Statistical power analysis.

**Figure supplement 5.** Summary-data-based Mendelian randomization.

**Figure supplement 6.** Ossification of the posterior longitudinal ligament of the spine (OPLL)-subtype stratification identified subtype-specific loci.

**Figure supplement 7.** A quantile–quantile plot of meta-analysis of subtype stratified genome-wide association studies.

**Figure supplement 8.** Regional association plots for three susceptibility loci for cervical ossification of the posterior longitudinal ligament of the spine (OPLL).

**Figure supplement 9.** Regional association plots for eight susceptibility loci for thoracic ossification of the posterior longitudinal ligament of the spine (OPLL).

**Figure supplement 10.** Expression levels of candidate genes in spinal ligament tissue in patients with cervical spondylotic myelopathy (CSM) and ossification of the posterior longitudinal ligament of the spine (OPLL).

**Figure supplement 11.** Expression levels of candidate genes in chondrogenic differentiated human ligament cells.

**Figure supplement 12.** Analyses of scRNA-seq GSE126060 data.

**Figure supplement 13.** Gene expression in each cluster at scRNA-seq GSE126060.

**Figure supplement 14.** Analyses of scRNA-seq GSE188758 data.

**Figure supplement 15.** Gene expression in each cluster at scRNA-seq GSE188758.

**Figure supplement 16.** Comparison of effect sizes of the genome-wide association studies (GWAS) lead single-nucleotide polymorphisms (SNPs) between the original GWAS meta-analysis and replication analysis.

**Figure supplement 17.** Comparison of effect sizes between ossification of the posterior longitudinal ligament of the spine (OPLL) and ankylosing spondylitis (AS) genome-wide association studies (GWASs) for AS-associated single-nucleotide polymorphisms (SNPs).

12p11.22 p=$1.39 \times 10^{-6}$) that reached locus-wide significance (p<$5.0 \times 10^{-6}$) (***Supplementary file 2***). Thus, the meta-analysis and conditional analysis identified 14 genome-wide significant OPLL loci, including 8 novel loci. Significant associations of the six previously reported loci (***Nakajima et al., 2014***) were observed in the present study (***Table 1***, ***Figure 1***, ***Figure 1—figure supplement 3***). The estimated proportion of the phenotypic variance explained by all the variants used in the study was

**Table 1.** Genome-wide significant loci in ossification of the posterior longitudinal ligament of the spine.

| SNP | CHR Position (Region start-end) | Gene | Novel/known | REF ALT | OPLL | p | OR (95% CI) | Phet† | GWAS 1 p | GWAS 1 ALT freq case | GWAS 1 ALT freq control | GWAS 1 OR (95% CI) | GWAS 2 p | GWAS 2 ALT freq case | GWAS 2 ALT freq control | GWAS 2 OR (95% CI) | GWAS 3 p | GWAS 3 ALT freq case | GWAS 3 ALT freq control | GWAS 3 OR (95% CI) |
|---|---|---|---|---|---|---|---|---|---|---|---|---|---|---|---|---|---|---|---|---|
| rs4665972 | 2 (p23.3) 27598097 (26598097–28598097) | SNX17 (intronic) | Novel | T C | ALL | 7.00E-09 | 1.23 (1.15–1.32) | 0.18 | 9.91E-07 | 0.483 | 0.433 | 1.27 (1.16–1.40) | 3.73E-04 | 0.474 | 0.425 | 1.26 (1.11–1.43) | 5.65E-01 | 0.441 | 0.430 | 1.05 (0.88–1.26) |
| | | | | | Cervical | 5.38E-05 | 1.25 (1.12–1.39) | 0.92 | 1.19E-03 | 0.481 | 0.433 | 1.25 (1.09–1.44) | 1.59E-02 | 0.469 | 0.425 | 1.24 (1.04–1.48) | – | • | | – |
| | | | | | Thoracic | 3.49E-02 | 1.14 (1.01–1.28) | 0.50 | 3.51E-02 | 0.478 | 0.433 | 1.24 (1.01–1.51) | 3.05E-01 | 0.454 | 0.425 | 1.16 (0.87–1.53) | 5.65E-01 | 0.441 | 0.430 | 1.05 (0.88–1.26) |
| rs927485 | 6 (p21.1) 44538139 (43529797–45538139) | LOC105375075, SUPT3H (intergenic) | Known | G A | ALL | 2.30E-09 | 0.76 (0.70–0.83) | 0.25 | 1.22E-07 | 0.824 | 0.864 | 0.72 (0.64–0.82) | 6.39E-02 | 0.843 | 0.860 | 0.86 (0.73–1.01) | 7.98E-03 | 0.829 | 0.872 | 0.74 (0.59–0.92) |
| | | | | | Cervical | 3.77E-03 | 0.82 (0.71–0.94) | 0.46 | 5.95E-03 | 0.835 | 0.864 | 0.79 (0.66–0.93) | 2.40E-01 | 0.846 | 0.860 | 0.87 (0.70–1.09) | – | | | – |
| | | | | | Thoracic | 7.48E-06 | 0.72 (0.62–0.83) | 0.92 | 2.63E-03 | 0.818 | 0.864 | 0.69 (0.55–0.88) | 4.16E-02 | 0.815 | 0.860 | 0.71 (0.51–0.99) | 7.98E-03 | 0.829 | 0.872 | 0.74 (0.59–0.92) |
| rs374810 | 8 (q23.1) 109096029 (108022775–110588327) | RSPO2 (upstream) | Known | G A | ALL | 1.03E-15 | 0.75 (0.70–0.81) | 0.93 | 9.56E-10 | 0.323 | 0.387 | 0.74 (0.68–0.82) | 2.72E-05 | 0.328 | 0.385 | 0.77 (0.68–0.87) | 2.06E-03 | 0.329 | 0.395 | 0.76 (0.64–0.90) |
| | | | | | Cervical | 6.04E-08 | 0.75 (0.67–0.83) | 0.14 | 6.95E-04 | 0.337 | 0.387 | 0.79 (0.69–0.91) | 7.42E-06 | 0.300 | 0.385 | 0.67 (0.56–0.80) | – | | | – |
| | | | | | Thoracic | 2.66E-07 | 0.73 (0.65–0.82) | 6.6E-02 | 2.81E-06 | 0.282 | 0.387 | 0.62 (0.50–0.75) | 4.85E-01 | 0.366 | 0.385 | 0.91 (0.70–1.19) | 2.06E-03 | 0.329 | 0.395 | 0.76 (0.64–0.90) |
| rs1898287 | 8 (q23.3) 117579970 (116484907–118588193) | LINC00536, EIF3H (intergenic) | Known | A C | ALL | 2.18E-10 | 0.80 (0.75–0.86) | 0.16 | 2.90E-09 | 0.605 | 0.668 | 0.75 (0.69–0.83) | 8.33E-03 | 0.625 | 0.664 | 0.85 (0.75–0.96) | 1.85E-01 | 0.633 | 0.664 | 0.89 (0.74–1.06) |
| | | | | | Cervical | 1.10E-02 | 0.87 (0.78–0.97) | 0.51 | 1.61E-02 | 0.633 | 0.668 | 0.85 (0.74–0.97) | 2.92E-05 | 0.641 | 0.664 | 0.91 (0.77–1.08) | – | | | – |
| | | | | | Thoracic | 2.18E-04 | 0.80 (0.71–0.90) | 0.10 | 7.40E-05 | 0.584 | 0.668 | 0.68 (0.56–0.82) | 3.80E-01 | 0.637 | 0.664 | 0.88 (0.67–1.16) | 1.85E-01 | 0.633 | 0.664 | 0.89 (0.74–1.06) |
| rs35505248 | 11 (q14.2) 86830927 (85724086–87887931) | TMEM135 (intronic) | Novel | T A | ALL | 6.75E-10 | 0.81 (0.75–0.86) | 0.44 | 1.76E-04 | 0.624 | 0.665 | 0.84 (0.76–0.92) | 7.06E-06 | 0.594 | 0.659 | 0.76 (0.67–0.85) | 1.90E-02 | 0.604 | 0.649 | 0.81 (0.68–0.97) |
| | | | | | Cervical | 1.06E-04 | 0.81 (0.73–0.90) | 2.7E-02 | 1.03E-01 | 0.640 | 0.665 | 0.89 (0.78–1.02) | 3.30E-05 | 0.577 | 0.659 | 0.70 (0.60–0.83) | – | | | – |
| | | | | | Thoracic | 4.53E-04 | 0.81 (0.72–0.91) | 0.65 | 7.62E-03 | 0.605 | 0.665 | 0.77 (0.64–0.93) | 4.67E-01 | 0.635 | 0.659 | 0.90 (0.69–1.19) | 1.90E-02 | 0.604 | 0.649 | 0.81 (0.68–0.97) |
| rs352811060 | 12 (p12.3) 19976182 (18955794–20077000) | AEBP2, LINC02398 (intergenic) | Novel | TG T | ALL | 1.39E-12 | 0.79 (0.74–0.84) | 0.58 | 3.50E-06 | 0.451 | 0.500 | 0.81 (0.74–0.88) | 2.92E-05 | 0.451 | 0.506 | 0.77 (0.69–0.87) | 4.50E-04 | 0.429 | 0.505 | 0.73 (0.61–0.87) |
| | | | | | Cervical | 1.06E-05 | 0.80 (0.72–0.88) | 0.43 | 2.74E-03 | 0.456 | 0.500 | 0.82 (0.72–0.93) | 8.81E-04 | 0.445 | 0.506 | 0.76 (0.64–0.89) | – | | | – |
| | | | | | Thoracic | 1.48E-06 | 0.75 (0.67–0.85) | 0.38 | 5.18E-04 | 0.424 | 0.500 | 0.72 (0.60–0.87) | 3.94E-01 | 0.482 | 0.506 | 0.89 (0.69–1.16) | 4.50E-04 | 0.429 | 0.505 | 0.73 (0.61–0.87) |
| rs10841442 | 12 (p12.2) 20213600 (20077000–21247540) | LINC02398 (ncRNA_intronic) | Known | T C | ALL | 1.03E-12 | 0.78 (0.73–0.84) | 0.61 | 1.07E-08 | 0.422 | 0.489 | 0.77 (0.70–0.84) | 6.60E-05 | 0.424 | 0.480 | 0.78 (0.69–0.88) | 7.56E-02 | 0.418 | 0.456 | 0.85 (0.71–1.02) |
| | | | | | Cervical | 1.57E-08 | 0.74 (0.67–0.82) | 0.71 | 2.87E-06 | 0.413 | 0.489 | 0.73 (0.64–0.83) | 1.40E-03 | 0.420 | 0.480 | 0.76 (0.65–0.90) | – | | | – |
| | | | | | Thoracic | 1.80E-04 | 0.80 (0.71–0.90) | 0.60 | 1.91E-03 | 0.417 | 0.489 | 0.74 (0.62–0.90) | 1.32E-01 | 0.432 | 0.480 | 0.82 (0.62–1.06) | 7.56E-02 | 0.418 | 0.456 | 0.85 (0.71–1.02) |

*Table 1 continued on next page*

*Table 1 continued*

| SNP | CHR Position (Region start-end) | Gene | Novel/known | REF ALT | OPLL | p | OR (95% CI) | $P_{het}$† | GWAS 1 ALT freq. case control | GWAS 1 p | GWAS 1 OR (95% CI) | GWAS 2 ALT freq. case control | GWAS 2 p | GWAS 2 OR (95% CI) | GWAS 3 ALT freq. case control | GWAS 3 p | GWAS 3 OR (95% CI) |
|---|---|---|---|---|---|---|---|---|---|---|---|---|---|---|---|---|---|
| rs11049529 | 12 (p11.22) 28471504 (27300776–28800000) | CCDC91 (intronic) | Known | C T | ALL | 1.01E-13 | 0.77 (0.72–0.83) | 0.63 | 0.569 0.629 | 6.72E-09 | 0.76 (0.69–0.83) | 0.564 0.627 | 1.31E-05 | 0.76 (0.67–0.86) | 0.572 0.601 | 5.63E-02 | 0.84 (0.70–1.00) |
| | | | | | Cervical | 2.57E-06 | 0.78 (0.70–0.87) | 0.89 | 0.575 0.629 | 3.06E-04 | 0.78 (0.69–0.90) | 0.566 0.627 | 2.55E-03 | 0.77 (0.65–0.91) | | - | - |
| | | | | | Thoracic | 9.93E-06 | 0.77 (0.68–0.86) | 0.29 | 0.541 0.629 | 6.68E-05 | 0.68 (0.57–0.82) | 0.577 0.627 | 1.15E-01 | 0.80 (0.61–1.05) | 0.572 0.601 | 5.63E-02 | 0.84 (0.70–1.00) |
| rs1038666 | 12 (p11.22) 29085005 (28800000–30107711) | CCDC91, FAR2 (intergenic) | Novel | G A | ALL | 5.09E-10 | 0.81 (0.76–0.87) | 0.06 | 0.573 0.609 | 1.43E-03 | 0.86 (0.79–0.95) | 0.532 0.613 | 8.18E-08 | 0.72 (0.64–0.81) | 0.553 0.601 | 2.03E-02 | 0.81 (0.68–0.97) |
| | | | | | Cervical | 5.48E-05 | 0.81 (0.74–0.90) | 0.29 | 0.569 0.609 | 1.12E-02 | 0.85 (0.75–0.96) | 0.546 0.613 | 9.33E-04 | 0.85 (0.75–0.96) | | - | - |
| | | | | | Thoracic | 2.89E-06 | 0.76 (0.68–0.85) | 0.22 | 0.551 0.609 | 9.54E-03 | 0.79 (0.65–0.94) | 0.496 0.613 | 3.43E-04 | 0.62 (0.48–0.81) | 0.553 0.601 | 2.03E-02 | 0.81 (0.68–0.97) |
| rs11157733 | 14 (q23.2) 50727523 (49727523–51729133) | L2HGDH (intronic) | Novel | G A | ALL | 2.65E-08 | 1.21 (1.13–1.29) | 0.58 | 0.463 0.423 | 2.90E-04 | 1.18 (1.08–1.30) | 0.478 0.419 | 7.52E-05 | 1.27 (1.13–1.43) | 0.460 0.426 | 7.18E-02 | 1.17 (0.99–1.38) |
| | | | | | Cervical | 5.28E-04 | 1.20 (1.08–1.32) | 0.74 | 0.461 0.423 | 1.20E-02 | 1.18 (1.04–1.34) | 0.468 0.419 | 1.60E-02 | 1.22 (1.04–1.44) | | - | - |
| | | | | | Thoracic | 1.48E-03 | 1.20 (1.07–1.34) | 0.19 | 0.446 0.423 | 2.53E-01 | 1.11 (0.93–1.34) | 0.517 0.419 | 2.89E-03 | 1.49 (1.15–1.93) | 0.460 0.426 | 7.18E-02 | 1.17 (0.99–1.38) |
| rs58255598 | 14 (q23.2) 62131805 (61131805–63131805) | FLJ22447, HIF1A-AS1 (intergenic) | Novel | C T | ALL | 2.16E-08 | 0.81 (0.75–0.87) | 0.76 | 0.276 0.319 | 1.75E-04 | 0.83 (0.75–0.91) | 0.278 0.324 | 1.67E-03 | 0.81 (0.71–0.92) | 0.272 0.324 | 4.88E-03 | 0.76 (0.63–0.92) |
| | | | | | Cervical | 2.19E-03 | 0.84 (0.75–0.94) | 0.36 | 0.287 0.319 | 6.19E-02 | 0.87 (0.76–1.01) | 0.271 0.324 | 9.53E-03 | 0.79 (0.65–0.94) | | - | - |
| | | | | | Thoracic | 1.36E-05 | 0.75 (0.66–0.86) | 0.96 | 0.254 0.319 | 4.37E-03 | 0.73 (0.59–0.91) | 0.270 0.324 | 8.52E-02 | 0.77 (0.57–1.04) | 0.272 0.324 | 4.88E-03 | 0.76 (0.63–0.92) |
| rs189646742 | 15 (q25.3) 88017055 (87017055–89017055) | AGBL1, LINC00052 (intergenic) | Novel | G A | ALL | 2.13E-08 | 2.03 (1.59–2.61) | 0.42 | 0.026 0.012 | 2.49E-07 | 2.31 (1.68–3.17) | 0.017 0.011 | 7.34E-02 | 1.57 (0.96–2.58) | 0.020 0.012 | 7.05E-02 | 1.85 (0.95–3.60) |
| | | | | | Cervical | 3.25E-05 | 2.14 (1.50–3.07) | 0.67 | 0.025 0.012 | 2.88E-04 | 2.27 (1.46–3.53) | 0.021 0.011 | 3.81E-02 | 1.92 (1.04–3.56) | | - | - |
| | | | | | Thoracic | 1.77E-02 | 1.72 (1.10–2.70) | 0.57 | 0.022 0.012 | 6.48E-02 | 1.88 (0.96–3.68) | 0.010 0.011 | 7.99E-01 | 0.83 (0.20–3.46) | 0.020 0.012 | 7.05E-02 | 1.85 (0.95–3.60) |
| rs376989376 | 16 (q22.1) 69854329 (68854329–70854329) | WWP2 (intronic) | Novel | T TAG | ALL | 1.08E-08 | 0.79 (0.73–0.86) | 0.45 | 0.660 0.693 | 4.48E-05 | 0.80 (0.71–0.89) | 0.677 0.702 | 1.28E-02 | 0.83 (0.72–0.96) | 0.639 0.699 | 7.28E-04 | 0.71 (0.58–0.87) |
| | | | | | Cervical | 2.70E-04 | 0.80 (0.71–0.90) | 0.83 | 0.658 0.693 | 2.65E-03 | 0.79 (0.68–0.92) | 0.673 0.702 | 3.87E-02 | 0.81 (0.66–0.99) | | - | - |
| | | | | | Thoracic | 4.10E-07 | 0.71 (0.62–0.81) | 0.81 | 0.631 0.693 | 5.18E-04 | 0.68 (0.54–0.84) | 0.663 0.702 | 1.07E-01 | 0.77 (0.56–1.06) | 0.639 0.699 | 7.28E-04 | 0.71 (0.58–0.87) |
| rs6140442 | 20 (p12.3) 7829397 (6713042–8882559) | MIR8062, HAO1 (intergenic) | Known | C A | ALL | 2.70E-14 | 1.39 (1.28–1.51) | 0.07 | 0.205 0.150 | 1.41E-11 | 1.48 (1.32–1.66) | 0.197 0.153 | 3.33E-05 | 1.38 (1.18–1.60) | 0.155 0.143 | 5.10E-01 | 1.08 (0.85–1.38) |
| | | | | | Cervical | 4.47E-08 | 1.42 (1.25–1.61) | 0.61 | 0.204 0.150 | 3.67E-06 | 1.46 (1.24–1.71) | 0.196 0.153 | 3.06E-03 | 1.36 (1.11–1.67) | | - | - |
| | | | | | Thoracic | 2.45E-02 | 1.19 (1.02–1.39) | 0.22 | 0.197 0.150 | 5.64E-03 | 1.39 (1.10–1.76) | 0.153 0.153 | 9.38E-01 | 1.01 (0.71–1.46) | 0.155 0.143 | 5.10E-01 | 1.08 (0.85–1.38) |

SNP single-nucleotide polymorphism; CHR, chromosome; REF, reference; ALT, alternative; OPLL, ossification of the posterior longitudinal ligament of the spine; GWAS, genome-wide association study; OR, odds ratio; CI, confidence interval; ALL, cervical + thoracic + others.

*Gene in or near region of association.

†Phet was derived from a Cochran's Q-test for heterogeneity.

53.1% (95% confidence interval [CI] 40.6–65.6%), indicating that OPLL has a high heritability. The lead variants of the 14 loci explained 6.5% of the phenotypic variance. Together with the LD score regression results, OPLL is a highly polygenic disease.

Adjacent to lead variants in the novel loci, we found several candidate genes (*Figure 1*) reported to be related to osteogenesis and could be connected to OPLL development. *TMEM135* (transmembrane protein 135), a gene in the newly identified significant locus (11q14.2), is a multi-transmembrane protein with seven transmembrane helices of high confidence. It is more strongly expressed in multipotent adipose tissue-derived stem cells committed to osteoblastic cells than the adipogenic lineage (*Scheideler et al., 2008*). *WWP2* (WW domain-containing E3 ubiquitin-protein ligase 2), the nearest gene to rs376989376 (the lead SNP in 16q22.1), was recently reported to serve as a positive regulator of osteogenesis by augmenting the transactivation of *RUNX2*, a master regulator of osteoblast differentiation as well as for chondrocyte maturation during skeletal development (*Zhu et al., 2017*).

All lead SNPs and SNPs in high LD ($r^2$ > 0.8) with them in previously unreported significant loci were in intron or intergenic regions, and none of them were exonic variants (*Supplementary file 3*). To prioritize putative causal variants, we conducted a Bayesian statistical fine-mapping analysis for significant loci using FINEMAP (*Benner et al., 2016*). The lead SNPs had the highest posterior probability (PP) in any significant region, and two of them were higher than 0.5: rs4665972 (2p23.3, p=0.548) and rs1038666 (12p11.22, p=0.533) (*Supplementary file 4*).

## Statistical power analysis

We examined the statistical power for minor allele frequency (MAF) and odds ratio of lead SNPs within the 14 independent significant regions in GWAS meta-analysis for ALL-OPLL. The results showed that

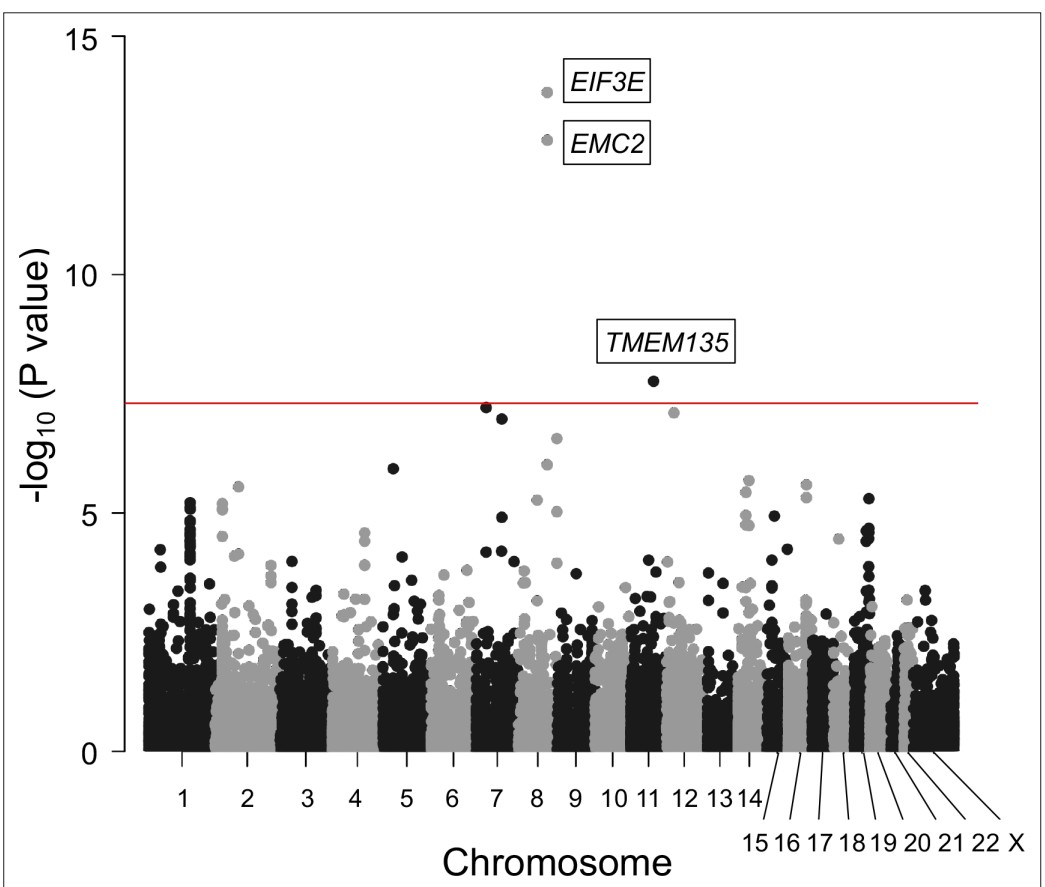

**Figure 2.** Gene-based association analysis identified five significantly associated genes in ossification of the posterior longitudinal ligament of the spine (OPLL). Manhattan plot showing the -log₁₀ p-value for each gene in the analysis. The values were plotted against the respective chromosomal positions. The horizontal red lines represent significance threshold (p=5.0 × 10⁻⁸).

all had a power greater than 0.5 for a significance level of p-value = $5 \times 10^{-8}$, and nine had a power greater than 0.8 (*Figure 1—figure supplement 4*).

## Enrichment in genes involved in bone metabolism

We conducted a gene set enrichment analysis implemented in FUMA (*Watanabe et al., 2017*). We found significant enrichment in the set related to bone mineral density (BMD): BMD of the heel (p=$8.60 \times 10^{-8}$), pediatric lower limb (p=$9.24 \times 10^{-5}$), and pediatric total body less head (p=$2.68 \times 10^{-4}$) (*Supplementary file 5*), compatible with the critical roles of bone metabolism in OPLL. However, we observed no significant enrichment in BMD in adults measured by dual-energy X-ray absorptiometry in this analysis.

## Identification of novel candidate genes missed by the GWAS meta-analysis

To identify other candidate genes, we conducted a gene-based association analysis (*de Leeuw et al., 2015*; *Watanabe et al., 2017*). We found three additional genes significantly associated with OPLL: *EIF3E*, *EMC2*, and *TMEM135* (*Figure 2*, *Supplementary file 6*). *EIF3E* and *EMC2* are in the same locus most strongly associated with OPLL as *RSPO2* (8q23.1.). *EIF3E* (eukaryotic translation initiation factor 3 subunit E) encodes a protein that is a component of the eukaryotic translation initiation factor 3 (eIF-3) complex, which functions in and is essential for several steps in the initiation of protein synthesis (*Lee et al., 2015*; *Masutani et al., 2007*). A proteomics study in a rat model of heterotopic ossification reported that Eif3e was upregulated in ossified tissues and may be involved in tissue ossification by regulating hypoxia-inducible factor (HIF) signaling, which has an important role in osteogenesis (*Wei et al., 2022*). *EMC2* (endoplasmic reticulum membrane protein complex subunit 2) encodes a part of the endoplasmic reticulum membrane protein complex (EMC) that functions in the energy-independent insertion of newly synthesized membrane proteins into the endoplasmic reticulum membrane, an essential cellular process (*Chitwood et al., 2018*; *O'Donnell et al., 2020*). However, basic experiments evaluating the effects of *EMC2* on ligament and bone tissue have not been reported, and the mechanisms involved in OPLL are unknown. On the other hand, this analysis reinforced the possible involvement of *TMEM135* in the development of OPLL.

The lack of exonic variants suggests that altering gene expression levels is a key function of OPLL-associated variants. By searching expression quantitative trait loci (eQTL) data in all available tissues in GTEx (*Consortium, 2015*), we found 26 transcripts with *cis*-eQTL variants associated with OPLL signals; of these, 20 transcripts were in the novel loci (*Supplementary file 7*). Furthermore, SMR (*Zhu et al., 2016*) revealed a total of 10 gene–tissue pairs (three unique genes, namely, *RSPO2*, *PLEC*, and *RP11-967K21.1*) that surpassed the genome-wide significance level ($P_{SMR} < 8.4 \times 10^{-6}$) without heterogeneity ($P_{HEIDI} < 0.05$) (*Supplementary file 8*). *RSPO2* is located in the most significant locus in GWAS meta-analysis, and its functions related to OPLL were elucidated in a past study (*Nakajima et al., 2016*). *PLEC* is expressed in various tissues, including muscles and fibroblasts (*Consortium, 2015*), and *PLEC* deficiency causes epidermolysis bullosa simplex with muscular dystrophy (OMIM 226670) (*Smith et al., 1996*), in which osteoporosis frequently develops (*Chen et al., 2019*). Since increased expression of *PLEC* was estimated to have a causal effect on OPLL (*Figure 1—figure supplement 5*, *Supplementary file 8*), these results suggest that *PLEC* is a likely causal gene of OPLL. As for *RP11-967K21.1*, its function in OPLL development is currently unknown and is expected to be elucidated in future studies.

## Cell groups and cell types related to OPLL

We conducted partitioning heritability enrichment analyses to investigate cell groups related to OPLL. We observed significant enrichment in the active enhancers of the connective/bone cell group and the immune/hematopoietic cell group (*Supplementary file 9*). We then analyzed each cell type belonging to these groups and found significant enrichment of H3K27ac in chondrogenic differentiation cells (*Supplementary file 10*). These results concord with previous findings that in OPLL chondrocyte differentiation in the endochondral ossification process occurs (*Sugita et al., 2013*) and provide new insights into the involvement of immune system cells in OPLL development, which has received little attention to date.

## Subtype analyses of OPLL

Subtype-stratified GWAS meta-analyses were also conducted: cervical (C)-OPLL (820 cases and 14,576 controls) and thoracic (T)-OPLL (651 cases and 20,007 controls). Subsequently, we identified three significant loci for C-OPLL and nine significant loci for T-OPLL (*Figure 1—figure supplements 6–9*, *Supplementary file 11*). Of these loci, one in the C-OPLL analysis and nine in the T-OPLL analysis were not identified in the analysis of ALL-OPLL and other OPLL subtypes. However, most of the lead SNPs in these significant loci were rare variants. We cannot determine that these are the causal variants based on the present results alone, but there was an interesting variant among them. rs74707424, a leading SNP in the significant locus (19p12), is located in the 3'-untranslated region of the ZBTB40 gene. In a recent study using primary osteoblasts of mouse calvaria, Doolittle et al. reported that *Zbtb40* functions as a regulator of osteoblast activity and bone mass, and knockdown of *Zbtb40*, but not *Wnt4*, in osteoblasts drastically reduced mineralization (*Doolittle et al., 2020*). We did not find significant genes in the gene-based analysis (data not shown).

## Expression of candidate genes in the spinal ligament in humans and mice

We then examined the expression of 23 genes of interest (ALL-OPLL GWAS: 19 genes; gene-based analysis: 2 genes; and SMR: 2 genes) in the spinal ligament, a target tissue of OPLL (see *Supplementary file 12* for detailed information on the number of genes). We used the deposited RNA-sequencing (RNA-seq) data in the spinal ligament (yellow ligament) in patients with OPLL and cervical spondylotic myelopathy (CSM) (GSE188760), and chondrogenic differentiation of human spinal ligament cells and controls (GSE188759) (*Tachibana et al., 2022*). Both data sets included 20/23 genes, of which we found 14/20 (70%) and 15/20 (75%) expressed in spinal ligament tissue in GSE188760 and GSE188759, respectively. In addition, the expression tended to be different between OPLL patients and CSM patients. Especially, the expressions of *WWP2*, *EIF3H*, and *SNX17* showed nominally significant differences (see 'Materials and methods' and *Figure 1—figure supplement 10*). *WWP2* was more highly expressed in chondrogenic differentiated ligament cells than in undifferentiated ligament cells (see 'Materials and methods' and *Figure 1—figure supplement 11*). The expression of *WWP2* has a positive effect on bone formation (*Scheideler et al., 2008*). We should revalidate these results with larger data sets in the future.

Next, to further explore expressions of these genes in single-cell levels, we used deposited single-cell RNA sequencing (scRNAseq) data of Achilles tendon cells in murine ossification models: burn/tenotomy heterotopic ossification model (GSE126060) (*Sorkin et al., 2020*) and Achilles tendon puncture model (GSE188758) (*Tachibana et al., 2022*). The Uniform Manifold Approximation and Projection (UMAP) identified 13 and 9 clusters in GSE126060 and GSE188758, respectively (*Figure 1—figure supplements 12–15*). Both data sets contained information on the same 14/23 genes. We confirmed that 12/14 (85.7%) of these genes are expressed in both mesenchymal and immune-related cells (macrophage, dendritic cell, and lymphocyte) in both data sets. These results are concordant with the results of partitioning heritability analysis, suggesting that not only the mesenchymal cells which differentiate into ligament and chondrocyte cells but also the immune cells are involved with ligament ossification.

We also conducted the same analyses for the candidate genes uniquely found in T- and C-OPLL and found the expression of most of the genes in ligamentous tissues.

## Causality of high BMI on OPLL

Epidemiological studies have suggested a relationship between OPLL and various other diseases and traits (*Akune et al., 2001*; *Endo et al., 2020*; *Kawaguchi et al., 2017*; *Kobashi et al., 2004*), particularly with T2D (*Akune et al., 2001*; *Kobashi et al., 2004*). We investigated their relationship with OPLL using the GWAS data. We first calculated the genetic correlation between OPLL and 96 complex traits (mean number of around 130K) (*Akiyama et al., 2019*; *Akiyama et al., 2017*; *Ishigaki et al., 2020*; *Kanai et al., 2018*; see *Supplementary file 13* for the traits analyzed). We found a positive genetic correlation between OPLL and BMI and T2D. The genetic correlation estimate (rg) was higher in the BMI group than in the T2D group. In addition, we identified new negative correlations between cerebral aneurysms (*Figure 3*, *Supplementary file 13*).

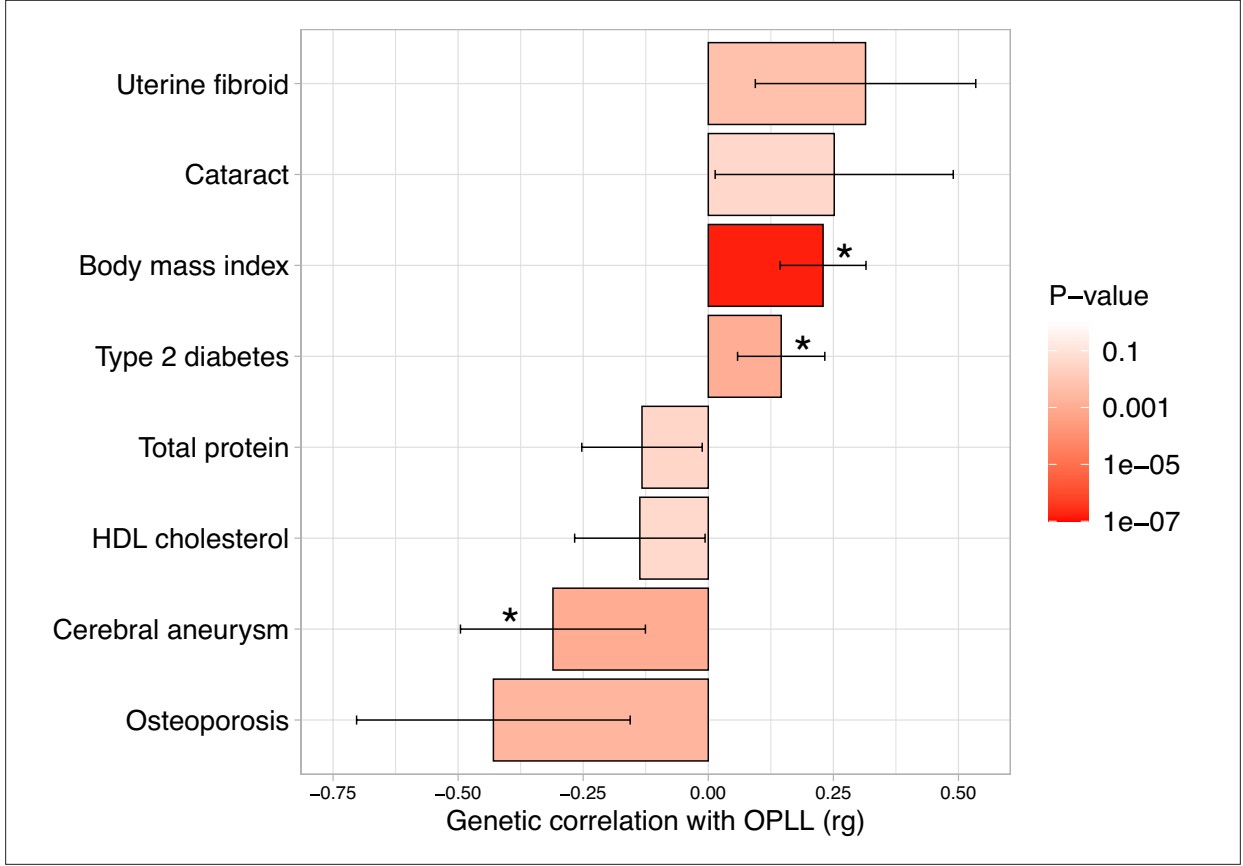

**Figure 3.** Genetic correlation between ossification of the posterior longitudinal ligament of the spine (OPLL) and other complex traits. Significant positive correlations with body mass index (BMI) and type 2 diabetes, and negative correlations with cerebral aneurysm were observed. Error bars indicate 95% confidence intervals. Red color gradations represent the level of p-value. Noted by asterisk is the significant correlation (false discovery rate [FDR] < 0.05).

Next, we conducted a two-sample MR using summary data from GWASs (*Akiyama et al., 2017*; *Bakker et al., 2020*; *Kemp et al., 2017*; *Spracklen et al., 2020*) to assess the causal effects of these significant traits in genetic correlation analysis on OPLL (*Evans and Davey Smith, 2015*; *Lawlor et al., 2008*; *Figure 4—figure supplement 1*, *Supplementary file 14*). The result of genetic correlation analysis for osteoporosis barely did not reach the false discovery rate (FDR)-corrected significance level, but we included it in the MR evaluation because there have been several previous reports of a strong trend toward whole-body ossification in OPLL patients (*Hukuda et al., 1983*; *Mori et al., 2016*; *Nishimura et al., 2018*; *Yoshii et al., 2019*). In the analysis, we used BMD, the main diagnostic criteria item for osteoporosis. BMD in the spine may reflect artifacts from OPLL itself, but higher BMD was also reported in patients with OPLL in the femur and the radius, a non-weight-bearing bone (*Sohn and Chung, 2013*; *Yamauchi et al., 1999*).

The significant causal effect of increased BMI on ALL-OPLL was estimated using the inverse variance weighted (IVW) method and the weighted median method (*Figure 4*, *Figure 4—figure supplement 2*, *Supplementary file 15*). The average pleiotropic effect of the MR-Egger regression intercept was close to zero (MR-Egger intercept = 0.005, p=0.581), indicating no evidence of the influence of directional pleiotropy (*Figure 4—figure supplement 2*, *Supplementary file 16*). We also assessed potential bias in the MR with a leave-one-out analysis and funnel plots; however, we did not identify any obvious bias (*Figure 4—figure supplement 3*). In contrast, we could not find any causal effects of T2D on ALL-OPLL with any MR methods (*Figure 4*, *Figure 4—figure supplement 4*, *Supplementary file 15*). As for BMD, we found a weak but significant causal effect of increased BMD on ALL-OPLL using multiple MR methods (*Figure 4*, *Figure 4—figure supplement 5*, *Supplementary file 15*), and the involvement of factors that stimulate bone formation in OPLL was suggested. Regarding cerebral aneurysms, the direction of the beta estimates differed according to the MR methods because of the

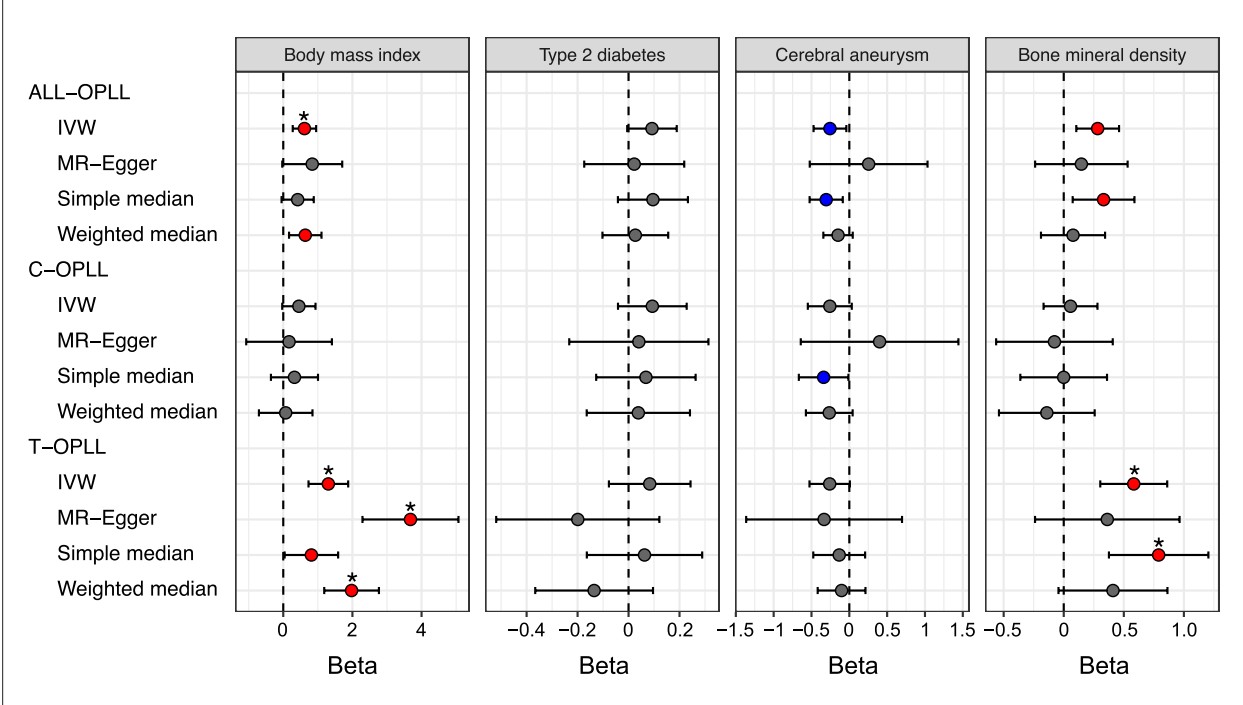

**Figure 4.** Causal effect of body mass index, type 2 diabetes, cerebral aneurysm, and bone mineral density on ossification of the posterior longitudinal ligament of the spine (OPLL). Causal effects were estimated using two-sample Mendelian randomization (MR) methods. Error bars indicate 95% confidence intervals. Significant (p<0.05) results are shown as red and blue dots for positive and negative causal effects, respectively. Noted by asterisk are the items that meet strict threshold (p<0.05/48=1.04 × 10$^{-3}$). IVW, inverse variance weighted.

The online version of this article includes the following figure supplement(s) for figure 4:

**Figure supplement 1.** Selection of single-nucleotide polymorphisms (SNPs) to be used as instrumental variables in Mendelian randomization.

**Figure supplement 2.** Scatter plots for the Mendelian randomization (MR) of the causal effect of body mass index (BMI) on ossification of the posterior longitudinal ligament of the spine (OPLL).

**Figure supplement 3.** Sensitivity analysis of the Mendelian randomization (MR) of body mass index (BMI) causality on ossification of the posterior longitudinal ligament of the spine (OPLL).

**Figure supplement 4.** Scatter plots for the Mendelian randomization (MR) of the causal effect of type 2 diabetes on ossification of the posterior longitudinal ligament of the spine (OPLL).

**Figure supplement 5.** Scatter plots for the Mendelian randomization (MR) of the causal effect of bone mineral density on ossification of the posterior longitudinal ligament of the spine (OPLL).

**Figure supplement 6.** Scatter plots for the Mendelian randomization (MR) of the causal effect of cerebral aneurysm on ossification of the posterior longitudinal ligament of the spine (OPLL).

**Figure supplement 7.** Scatter plots for the Mendelian randomization (MR) of the causal effect of ossification of the posterior longitudinal ligament of the spine (OPLL) on body mass index (BMI), type 2 diabetes, cerebral aneurysm, and bone mineral density.

**Figure supplement 8.** Scatter plots for the Mendelian randomization (MR) of the causal effect of body mass index (BMI) on ossification of the posterior longitudinal ligament of the spine (OPLL) subtypes.

**Figure supplement 9.** Sensitivity analysis of the Mendelian randomization (MR) of body mass index (BMI) causality on ossification of the posterior longitudinal ligament of the spine (OPLL) subtypes.

**Figure supplement 10.** Mendelian randomization (MR) for obesity-related traits on ossification of the posterior longitudinal ligament of the spine (OPLL).

**Figure supplement 11.** Correlation of the effect sizes of the genome-wide single-nucleotide polymorphisms (SNPs) of ossification of the posterior longitudinal ligament of the spine (OPLL) and body mass index (BMI).

**Figure supplement 12.** Replication of Mendelian randomization (MR) for body mass index, type 2 diabetes, cerebral aneurysm, and bone mineral density on ossification of the posterior longitudinal ligament of the spine (OPLL).

**Figure supplement 13.** Replication of Mendelian randomization (MR) for obesity-related traits on ossification of the posterior longitudinal ligament of the spine (OPLL).

small number of SNPs used as the instrumental variables (*Figure 4*, *Figure 4—figure supplement 6*, *Supplementary file 15*).

We performed a reverse-direction MR to evaluate the causality of OPLL on BMI, T2D, cerebral aneurysm, and BMD but did not find any significant causal effects on the four traits (*Figure 4—figure supplement 7*, *Supplementary files 17 and 18*).

## The large causal effect of high BMI and high BMD on T-OPLL

We estimated the causal effect of traits with a significant genetic correlation with OPLL subtypes. We found contrasting results between C- and T-OPLL. A significant causal effect of increased BMI on T-OPLL, but not on C-OPLL, was indicated by all four MR methods. All beta estimates on T-OPLL were greater than those in the analysis for ALL-OPLL (*Figure 4*, *Figure 4—figure supplements 2 and 8*, *Supplementary file 15*), suggesting that T-OPLL drove the causal effect of BMI on OPLL. The MR-Egger regression intercept was significantly negative, suggesting the existence of directional pleiotropy (*Figure 4—figure supplement 8*, *Supplementary file 16*); however, the results of other sensitivity analyses for robust causal inference (simple and weighted median methods) suggested its causality on T-OPLL (*Figure 4*, *Figure 4—figure supplements 8 and 9*, *Supplementary file 15*). As for BMD, a larger causal effect of increased BMD on T-OPLL compared to ALL-OPLL was also estimated (*Figure 4*, *Figure 4—figure supplement 5*, *Supplementary file 15*). We did not find any significant effects in T2D or cerebral aneurysms except for the results using the simple median method for cerebral aneurysms.

## Additional MR for obesity-related traits

Based on the above MR results, we focused on the causal relationship between high BMI, that is obesity, and OPLL. First, we repeated MR using only Japanese BMI data (*Akiyama et al., 2017*) to estimate causal effects on OPLL as a sensitivity analysis. As a result, we confirmed the positive direction of the effect. Furthermore, the results were significant for T-OPLL, reinforcing the causality of BMI on T-OPLL (*Supplementary file 19*).

Next, we conducted additional MR for obesity-related traits. We used the data from the large GWAS meta-analyses for obesity-related traits from UK Biobank (UKBB) and Giant Consortium: BMI, waist-to-hip ratio (WHR), and WHR adjusted by BMI (WHRadjBMI) (*Pulit et al., 2019*; *Supplementary file 20*). Regarding causality from BMI to OPLL, all four MR methods showed significant causality for ALL- and T-OPLL, while only IVW method showed causality for C-OPLL (*Figure 4—figure supplement 10*, *Supplementary file 21*). Regarding causality from WHR to OPLL, 2/4 and 3/4 MR methods showed significant causality for ALL- and T-OPLL, respectively, but no significant results were obtained for C-OPLL by either method. For both traits, the magnitude of the causal effect size estimated by MR tended to be larger for C-, ALL-, and T-OPLL, in that order, suggesting a strong influence of obesity on the development of T-OPLL. On the other hand, no significant causal relationship was found in the MR on OPLL from WHRadjBMI, a surrogate index of abdominal adiposity. It suggests that systemic obesity, rather than a simple high percentage of abdominal fat, influences the development of OPLL.

## The polygenic causal effect of high BMI on OPLL

In addition to the causal effects of significant variants with BMI on OPLL, we evaluated the shared polygenic architecture between BMI and OPLL. Moderate correlations were found between the effect sizes of the SNPs of ALL- and T-OPLL with BMI, especially in sets of SNPs with low p-values (p<0.0005), when calculating the correlation based on p-value in the BMI GWAS. We did not find such correlations when calculating the correlation based on the p-value in the OPLL GWAS (*Figure 4—figure supplement 11*). However, this difference was not observed in the sets of SNPs of any p-value groups in the C-OPLL analysis. These results support the theory that high BMI is one of the causal factors of OPLL, and its causal effect on OPLL is driven by that on T-OPLL.

## Heterogeneity of impact of obesity inside OPLL subtypes

Finally, we generated a PRS of BMI and compared its effect on OPLL for ALL-OPLL, C-OPLL, and T-OPLL (*Figure 5—figure supplements 1 and 2*). We found that BMI PRS could predict OPLL, especially T-OPLL (*Figure 5*). Further, the effect sizes of BMI PRS on OPLL were all larger than that on T2D

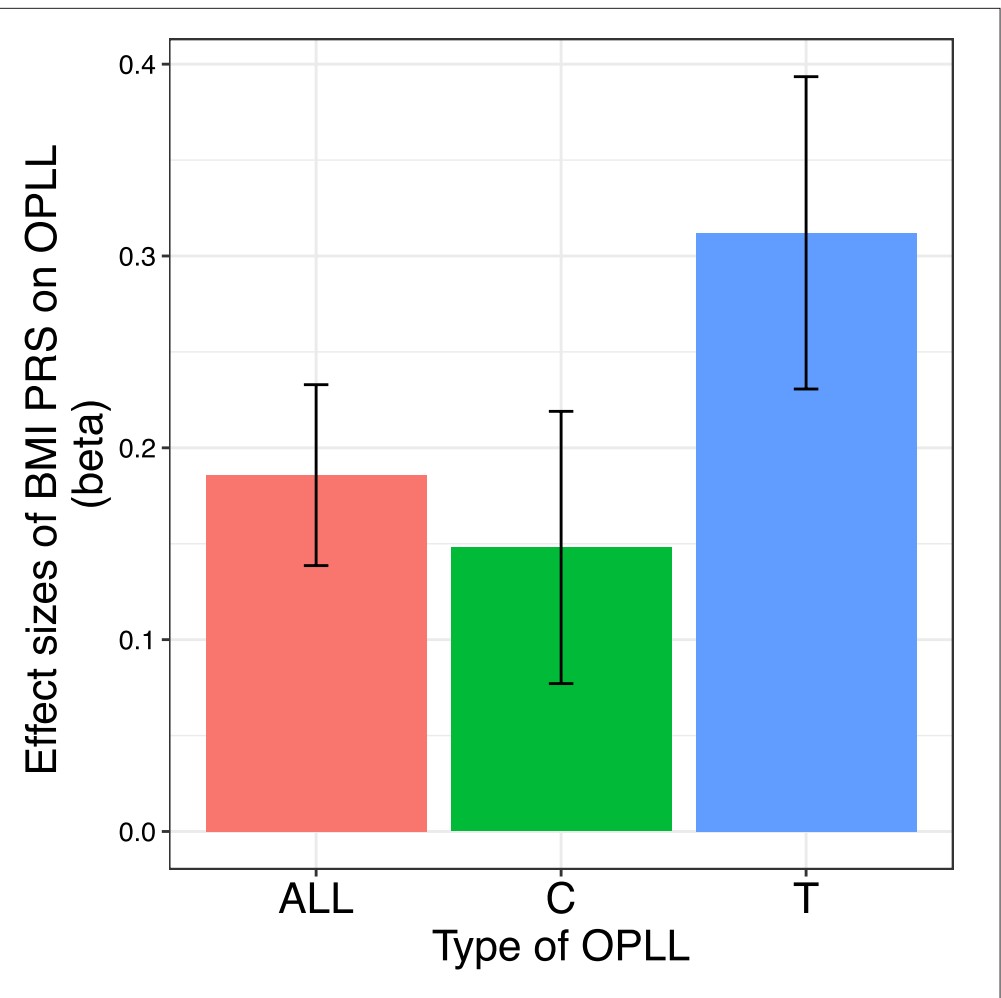

**Figure 5.** Body mass index (BMI) polygenic risk score predicts ossification of the posterior longitudinal ligament of the spine (OPLL). Vertical columns represent effect sizes of BMI polygenic risk score (PRS) on three types of OPLL: cervical (C-) OPLL, thoracic (T-) OPLL, and ALL-OPLL (C-OPLL, T-OPLL, and others). The BMI PRS could predict OPLL, especially T-OPLL. Error bars represent the 95% confidence intervals of the effects.

The online version of this article includes the following figure supplement(s) for figure 5:

**Figure supplement 1.** Body mass index (BMI) polygenic risk score analysis for ossification of the posterior longitudinal ligament of the spine (OPLL).

**Figure supplement 2.** Determination of the best parameter for body mass index (BMI) polygenic risk score.

---

analyzed in a similar manner ('Materials and methods') (beta = 0.099, 95% CI 0.087–0.110), indicating that high BMI is a major cause of OPLL.

Comparing C- and T-OPLL, we found that the effect of BMI PRS was significantly greater in T-OPLL (p=0.016), even in a limited number of cases.

## Validity of this study reinforced by the replication study

To assess the validity of our GWAS meta-analysis, we conducted an association analysis using replication data of additional 2105 individuals (212 T-OPLL cases and 1866 controls). As for ALL-OPLL, we confirmed that the direction of betas for the lead SNPs in most regions (13/14) is consistent between the GWAS meta-analysis and the replication study (p=$1.83 \times 10^{-3}$, binomial test). Thus, the validity of the ALL-OPLL signals was reinforced (*Figure 1—figure supplement 16*). As for T-OPLL, the concordant rate was a little lower (6/9 of the lead SNPs), explained by low power in the additional data set to detect associations of rare alleles.

Next, we re-analyzed MR with expanded results of OPLL ('Materials and methods'). The addition of new replication data further strengthened the significance of the causal effect from BMI and BMD to OPLL (*Figure 4—figure supplement 12*, *Supplementary files 15 and 16*). For T2D, the result was only nominally significant in the IVW method (p=4.86 × 10⁻²), and the causal effect from T2D to OPLL was not certain. Results of cerebral aneurysms showed little change, indicating a little causal effect on OPLL. We performed the same replication analysis for MR using obesity-related traits data from UKBB and Giant consortium (*Figure 4—figure supplement 13*, *Supplementary files 21 and 22*). The causal relationship from them to OPLL was further reinforced for BMI and WHR. WHRadjBMI demonstrated comparable results.

## Discussion

This study conducted a GWAS meta-analysis using 2010 OPLL cases and 20,006 controls and identified 14 significant loci, including 8 novel loci. Although many unknowns exist regarding the function of the six regions found in the past, functional analysis of chromosome 8 (q23.1), where *RSPO2* is located, has progressed. rs374810, the most significant variant in the present and the past study, is located on the promoter region of *RSPO2* in chondrocytes. Its risk allele (C allele) causes a loss of transcription factor C/EBPβ binding; therefore, *RSPO2* expression is reduced. Finally, the Wnt-β-catenin signal that blocks chondrocyte differentiation of ligament MSCs is suppressed, which triggers OPLL generation (*Nakajima et al., 2016*). Some of the newly discovered OPLL-related signals are in or near genes such as *TMEM135* and *WWP2,* which are associated with the activation of bone formation (*Scheideler et al., 2008*; *Zhu et al., 2017*). In addition, we identified six genes associated with OPLL using gene-based analysis and an SMR: *EIF3E, EMC2, TMEM135, PLEC, RSPO2*, and *RP11-967K21.1*. *RSPO2* and *TMEM135* were also found in the GWAS meta-analysis, reinforcing the results. *EIF3E* is suspected to be involved in heterotopic ossification by regulating HIF signaling, which has an important role in osteogenesis (*Wei et al., 2022*). In addition, *EIF3E* is also the nearest gene to the lead variant within the significant locus in previous GWAS for heel bone mineral density: rs7815105; beta = –0.014 (same direction as this study); p=4.9 × 10⁻¹¹ (*Morris et al., 2019*). The *PLEC* mutation causes epidermolysis bullosa, in which osteoporosis is one of the main comorbidities. Although various causes have been reported to induce osteoporosis in this disease (*Chen et al., 2019*), the underlying biological mechanism by which *PLEC* mutations affect bone metabolism is unclear, and future studies are expected to elucidate this mechanism. Although the effects of *EMC2* and *RP11-967K21.1* on ligament and bone tissue have not been reported and are unknown, future studies may elucidate the mechanisms and reveal that they are the actual causal genes. Thus, our GWAS observations are compatible with the theory that OPLL is closely related to bone metabolism and develops through endochondral ossification (*Sato et al., 2007*; *Sugita et al., 2013*).

Partitioning heritability enrichment analyses by LD score regression identified that OPLL GWAS signals enrich not only the active enhancers of the connective/bone cell group but also those of the immune/hematopoietic cell group. It is well known that heterotopic ossification of ligaments and tendons, in which mesenchymal stem cells differentiate into osteochondrogenic cells rather than myocytes or tenocytes, is triggered by tissue damage due to trauma (*Convente et al., 2018*; *Torossian et al., 2017*). Tissue injury triggers a systematic inflammatory response with the mobilization of neutrophils, monocytes, and lymphocytes at various stages of inflammation. Monocytes/macrophages are involved in abnormal wound healing and influence the development of heterotopic ossification (*Sorkin et al., 2020*). This fact suggests that immune system cells are involved in the tissue repair process of the posterior longitudinal ligament tissue of the spine, which is the host that causes ossification in OPLL as well. In addition, gene expression analysis using scRNAseq data confirmed that the candidate genes discovered in this study were expressed not only in mesenchymal cells but also in immune cells.

We identified that OPLL was genetically correlated with other traits, positive for BMI and T2D and negative for cerebral aneurysms. High BMI (obesity) has been implicated in OPLL, and clinical studies have reported that the BMI of OPLL patients is significantly higher than that of non-patients (*Endo et al., 2020*; *Kobashi et al., 2004*). It is speculated that obesity promotes heterotopic bone formation in OPLL. Yamamoto et al. reported a high incidence of OPLL in the Zucker fatty rat, a model rat of obesity with a loss-of-function mutation in the leptin receptor gene (*Yamamoto et al., 2004*). However, basic studies with small animals have reported conflicting results regarding the

effects of obesity on bone formation. In general, high-fat diet (HFD)-induced obesity is detrimental to bone formation and results in low bone density in mice (*Zhang et al., 2022*). On the other hand, Lv et al. reported that the mRNA level of *Runx2* in bone marrow-derived mesenchymal stem cells from HFD mice was significantly higher; in contrast, that of *Ppary*, which suppresses osteoblast differentiation and promotes osteoclast differentiation, was significantly lower than the control mice (*Lv et al., 2010*). Thus, the results of basic experiments are varied and controversial. While the effects of obesity on systemic bone formation and heterotopic ossification may not be the same, in clinical studies, meta-analyses examining the association between obesity and BMD in adults have reported that obesity affects the increase in bone density in all groups: postmenopausal women, premenopausal women, and men. In addition, MR analysis of the relationship between BMI and BMD reported that increased BMI was positively causally related to heel BMD (*Ma et al., 2021*; *Song et al., 2020*) and lumbar BMD (*Song et al., 2020*). Thus, while many results support the theory that obesity is related to OPLL formation, the mechanism by which obesity induces OPLL remains unclear. Our MR studies have demonstrated that a high BMI has a causal effect on OPLL. However, we could not prove the SNP-obesity interaction (Appendix 2). Further mechanistic studies on obesity and OPLL are necessary.

We found that the causal relationship between T2D and OPLL is slight by our MR analysis, despite the positive genetic correlation between OPLL and T2D. Many OPLL studies have focused on the relationship with T2D (*Akune et al., 2001*; *Kobashi et al., 2004*). Insulin acts on endogenous tyrosine kinase receptors and receptors for insulin-like growth factor-I, a potent anabolic factor in bone formation (*Giustina et al., 2008*; *Locatelli and Bianchi, 2014*). Therefore, it has been postulated that increased insulin production due to impaired insulin action may stimulate osteogenic cells in ligaments and cause OPLL (*Akune et al., 2001*). However, our results indicate that the impact of T2D on the development of OPLL is small. Therefore, most reasons for the high prevalence of T2D in OPLL patients reported so far can be attributed to the high prevalence of obesity in OPLL patients.

Our MR analysis also revealed the causality of high BMD in OPLL. This result is consistent with the clinical observation that OPLL patients have an increased tendency of ossification throughout the body and often have heterotopic ossification of other spinal ligaments, such as the anterior longitudinal ligament (*Nishimura et al., 2018*), interspinous ligament (*Mori et al., 2016*), and nuchal ligament (*Yoshii et al., 2019*), as well as extraspinal ossification lesions in the shoulder, hip, knee, and ankle joint (*Hukuda et al., 1983*). A gene set enrichment analysis identified significant enrichment in sets associated with BMD in children as well as adults. BMD is low at birth, increases gradually with age, peaks in the 20 s, and then slowly declines (*Iglesias-Linares et al., 2016*). Therefore, it is assumed that BMD in children reflects the degree of bone formation, although it is affected by various factors. Our results can support that OPLL is closely associated with bone formation. However, the relationship between cerebral aneurysms and OPLL has not been reported. Therefore, for evaluating the causality of cerebral aneurysms on OPLL, an MR analysis with more cerebral aneurysm-related SNPs as the instrumental variables is desirable. Such a study will be conducted in the future. In addition to these traits, additional analyses were performed for ankylosing spondylitis with ossifying lesions in the spine as well as OPLL. We evaluated the correlation between ankylosing spondylitis and OPLL but found no correlation in the limited data available (Appendix 3, *Figure 1—figure supplement 17*).

We demonstrated the differences in genetic characteristics of OPLL. We identified three significant loci for C-OPLL and nine significant loci for T-OPLL in the stratification analyses of the GWAS by OPLL subtypes. There was a considerable difference in the allele frequencies of lead variants in these loci between subtypes (*Supplementary file 11*). However, most of the alternative allele frequencies of these variants were small, and future confirmation by a study with a larger sample size is desirable. Furthermore, our MR and BMI PRS analyses showed that the effect of BMI on T-OPLL was much larger than that of C- and ALL-OPLL. The common disease study often shows clinically defined diseases based on common signs and symptoms are actually heterogeneous in the cause, such as hypertension and diabetes mellitus. Therefore, research focusing on disease subtypes is useful in characterizing the disease in detail and elucidating its specific causes, leading to more personalized treatment.

This study demonstrates that OPLL is a highly heritable disease. Although previous clinical studies have suggested that OPLL is heritable, there have been no studies that have mentioned the heritability of OPLL. This study is the first to clarify the high heritability of OPLL quantitatively. Further elucidation of the pathogenesis of OPLL from a genetic approach is expected.

Based on the results of this study, we expect to establish new non-surgical treatments. Although this study identified that OPLL is closely linked to bone metabolism, there are few studies examining the prognosis of OPLL with the use of bone-modifying agents such as bisphosphonates, which suppress both bone formation and resorption. A randomized controlled trial evaluated the effect of etidronate disodium on postoperative OPLL progression in patients with OPLL who had undergone posterior decompression surgery, but no significant effect was demonstrated (*Yonenobu et al., 2006*). Future studies are needed to determine whether existing agents are adequate or whether we need to develop new agents. Regarding obesity, the cause of OPLL, weight loss guidance, and aggressive bariatric treatment, including surgery, for some patients may be considered. Long-term prospective studies are needed to evaluate the effect of weight loss on OPLL suppression. In addition, this study suggests an involvement of the immune system in OPLL. Although most OPLL research to date has focused on bone metabolism, it is also essential to study the pathogenesis of OPLL from an immunological approach in the future to develop unprecedented therapies.

There are some limitations in this study. The first is the GWAS sample size. Although this study is the largest OPLL GWAS in the world and used samples collected from facilities throughout Japan, additional samples are needed to clarify the pathogenesis of OPLL further and better define subtype differences. In addition, OPLL GWAS in other ethnic groups has not been reported. We expect future international meta-analysis using data from non-Japanese ethnic groups to elucidate regional differences in the frequencies of OPLL. Second, gene expression data in spinal ligament tissue is scarce. Therefore, we were forced to use tissue data different from ligament tissue, such as GETx (*Consortium, 2015*), which did not include bone, cartilage, or ligamentous tissue, and data from a mouse Achilles tendon ossification model, which resembles spinal ligament ossification. We also used bulk data from spinal ligaments, but the sample sizes were limited. We focused on the genes closest to the GWAS signals, but some of these genes may not be the true causative genes because their expression was not confirmed or low in ligament tissues as far as we could ascertain from the limited data available. It is desirable to increase the number of spinal ligament tissue samples and perform expression analysis using scRNAseq. Additionally, we expect that functional data such as eQTL in ligament tissues or other responsible tissues will be available in the future. If so, genes other than those focused on in this study could be identified as causal genes. Another limitation is that the Japanese GWAS data used for genetic correlation analysis were limited to 96 traits, and other traits not analyzed for genetic correlation may be associated with OPLL. In addition, some of the data used for MR were not East Asian. While we cannot immediately improve upon the limitations listed above, we intend to strengthen these areas for future research for OPLL.

In conclusion, this study identified candidate genes in genomic loci associated with OPLL, and subsequent post-GWAS analyses showed a causal relationship between other traits and OPLL: obesity and high BMD. This study will serve as a basis for future research to elucidate the pathogenesis of OPLL in more detail and to develop new treatment methods.

## Materials and methods
### Subjects

All the subjects analyzed in this study were Japanese. OPLL samples were collected from facilities throughout Japan. The GWAS data of this study consisted of three sets: GWAS set-1, -2, and -3. The case samples of set-1 and -2 were used as discovery and replication samples, respectively, in the previous GWAS (*Nakajima et al., 2014*). In these data sets, the cases had OPLL of more than or equal to two vertebrae. For the case of set-3, we recruited patients with OPLL in more than or equal to five vertebras or OPLL thicker than 5 mm in the thoracic spine in 2018–2019. When assessing the presence or absence of OPLL, expert spine surgeons in each institution examined patients' plain radiography or computed tomography (CT) in detail (*Figure 1—figure supplement 1*).

Regarding control data, we used genotyping data from BioBank Japan (BBJ) (*Hirata et al., 2017*; *Nagai et al., 2017*) in set-1 and -2, and those from the Medical Genome Center (MGC) Biobank database of the National Center for Geriatrics and Gerontology (NCGG) in set-3. Details of the characteristics of the subjects are shown in *Supplementary file 1*.

This study followed the Strengthening the Reporting of Genetic Association Studies (STREGA) reporting guideline (*Little et al., 2009*).

## Study approval

All participating individuals provided written informed consent to participate in this study following approval by ethical committees at RIKEN Centers for Integrative Medical Sciences (approval ID: 17-16-39), Hokkaido University (approval ID: 16-059), and all other participating institutes.

## Genotyping and quality control

Genomic DNA was extracted from peripheral venous blood samples using a standard method. We genotyped case and control samples using the Illumina OmniExpressExome BeadChip, a combination of Illumina OmniExpress BeadChip and Illumina HumanExome BeadChip, or Illumina Asian Screening Array (*Supplementary file 1*).

For quality control of genotyped SNPs, we excluded those with (i) SNP call rate < 99%, (ii) MAF < 0.01, and (iii) Hardy–Weinberg equilibrium p-value<$1.00 \times 10^{-6}$. We constructed a reference panel to obtain imputed genotypes with high *accuracy* using the 1000 Genomes Project Phase 3 (1KGP 3 [May 2013 n=2504]) and 3256 in-house Japanese whole-genome sequence data obtained from BBJ (JEWEL 3K) in the same way as previously reported (*Akiyama et al., 2019*). SNPs with allele frequency differences greater than 0.06 between the genotyped control data and the 1KGP3 East Asian and JEWEL 3K data in reference panel were excluded.

For sample quality control, we excluded samples whose sex differed between genotype and clinical data. We evaluated cryptic relatedness by calculating estimates of pairwise IBD (PI_HAT) and removed samples that showed second-degree relatedness or closer (PI_HAT > 0.25). Population stratification was estimated using principal component analysis (PCA) with four populations from HapMap data as the reference: European (CEU), African (YRI), Japanese (JPT), and Han Chinese (CHB) with SmartPCA (*Patterson et al., 2006*). We generated a scatterplot using the top two associated principal components (eigenvectors) and selected samples within the East Asian (JPT/CHB) cluster. We excluded samples with a genotyping call rate of <98% (*Figure 1—figure supplement 1*).

## Phasing and genotype imputation

We performed pre-phasing with EAGLE (v2.4.1) (*Loh et al., 2016*) and SNP imputation with minimac4 (v1.0.0) (*Das et al., 2016*) using the reference panel mentioned above. After imputation, we used SNPs with an imputation quality of Rsq > 0.3 and MAF > 0.005 for the subsequent association study.

## GWAS and meta-analysis

We performed an association analysis of autosomes of GWAS set-1, -2, and -3 independently. We performed a logistic regression analysis using PLINK2.0 (*Purcell et al., 2007*) with the top 10 principal components (PCs) as covariates assuming an additive model, and evaluated the association of each imputed SNP. We then meta-analyzed the three GWAS sets with an inverse variance method under a fixed effect model using METAL software (*Willer et al., 2010*). Regarding X chromosomes, we performed a logistic regression analysis in males and females separately for each GWAS set using PLINK2.0, with the top 10 PCs as covariates assuming an additive model. We then integrated the results of males and females in each GWAS using an inverse variance method under a fixed-effect model (*Figure 1—figure supplement 1*).

We estimated confounding biases derived from population stratification and cryptic relatedness using LD score regression using LD scores for the East Asian population (*Bulik-Sullivan et al., 2015*).

## Conditional association analysis

We used the distance-based approach to determine significant loci. We defined the SNP with the lowest p-value within each locus as the lead SNP. We defined an associated locus of a lead SNP as 1 Mb of its surrounding sequences in both directions. We extended the region to nearby significant variants and their 1 Mb surrounding sequences as far as a significant variant was contained in the defined region. In addition, we margined 1 Mb from significant variants at both ends. We performed a stepwise conditional meta-analysis to determine the independent association signals in the associated loci. We conducted conditional analyses of GWAS set 1–3 separately and integrated the results using a fixed-effects model with the inverse variance method. We repeated this process until the top associated variants fell below the locus-wide significance level (p<$5.0 \times 10^{-6}$) in each stepwise procedure. As a result of this analysis, we identified two additional loci associated with OPLL (lead SNPs:

rs35281060 and rs1038666). We determined the boundaries of these regions based on the estimated recombination rates from hg19/1000 Genomes Nov 2014 East Asian (*Table 1*). We confirmed that there are no SNPs outside each locus in LD ($r^2 > 0.1$) with SNPs that met genome-wide significance levels within the locus.

We calculated LD with the lead SNPs using whole-genome sequence data of 1KGP3 East Asian and JEWEL 3K by PLINK2.0 and produced regional association plots using Locuszoom (*Pruim et al., 2010*).

## Statistical power analysis

We evaluated the statistical power of GWAS meta-analysis for ALL-OPLL using the genpwr package for R (*Moore et al., 2019*). We set the model to an additive model, with the type 1 error rate and statistical power set to $5 \times 10^{-8}$ and 0.8 or 0.5, respectively.

## Annotation

We used ANNOVAR (*Wang et al., 2010*) and defined a gene with a lead SNP or, if not, a gene in the nearest vicinity of a lead SNP as a candidate gene for the region because of previous reports that the majority of noncording variants act on the closest gene (*Fulco et al., 2019*; *Nasser et al., 2021*).

To identify candidate causal variants in the eight novel loci, we annotated the SNPs that exceeded the threshold of significance ($p<5.0 \times 10^{-8}$) and were in high LD ($r^2 > 0.8$) with lead variants newly identified in the GWAS meta-analysis. We explored the biological role of these variants using SNP annotation tools, including HaploReg (*Ward and Kellis, 2012*), RegulomeDB (*Boyle et al., 2012*), and ANNOVAR (*Wang et al., 2010*).

## Subtype-stratified GWAS and meta-analysis

We performed subtype-stratified GWASs and meta-analyses for C-OPLL and T-OPLL in the same way as the analysis with all samples from set to 1–3 (ALL-OPLL). Cases in GWAS set-3 were all T-OPLL samples; therefore, we carefully re-examined patients' plain radiography or CT in set-1 and -2, where we defined C-OPLL case samples as those with OPLL limited to the cervical spine and defined T-OPLL as OPLL affecting more than two vertebrae in the thoracic spine (*Figure 1—figure supplement 1*). The detailed sample numbers are listed in *Supplementary file 1*.

## Estimation of phenotypic variance

We estimated the heritability of OPLL using LDSC software (*Bulik-Sullivan et al., 2015*). The variance explained by the variants was calculated based on a liability threshold model assuming the prevalence of OPLL to be 3.0% (*Matsunaga and Sakou, 2012*). Furthermore, the model assumed that subjects had a continuous risk score and that subjects whose scores exceeded a certain threshold developed OPLL.

## Bayesian statistical fine-mapping analysis

To prioritize causal variants in OPLL susceptibility loci, we conducted a fine-mapping analysis using FINEMAP v1.3 software (*Benner et al., 2016*), using z-scores of GWAS meta-analysis for ALL-OPLL and LD matrices calculated by 1KGP3 EAS and JEWEL 3K data. We assumed one causal signal in the ±1 Mb region from both ends of significant variants at each significant locus. However, for 12p11 and 12p12, in which we identified significant secondary signals by a conditional analysis, we defined the range of the region referring to regional association plots (*Figure 1—figure supplement 3*). We calculated a PP in which each genetic variant was the true causal variant. Then, we ranked the candidate causal variants in descending order of their PPs and created a 95% credible set of causal variants by adding the PPs of the ordered variants until their cumulative PP reached 0.95.

## Gene set enrichment analysis

FUMA is a web-based platform in which we can perform GWAS-related analyses such as gene-based analysis and gene set enrichment analysis (*Watanabe et al., 2017*). We conducted a gene set enrichment analysis using FUMA. Because variants often act on genes that are close in the distance (*Fulco et al., 2019*; *Nasser et al., 2021*), we selected genes on a distance basis using the following criteria

and used them as input data: genes (i) located within 1 Mb and (ii) the five closest to the leading SNPs of each genome-wide significant locus.

## Gene-based association analysis

To examine the combined effect of SNPs, we conducted gene-based association analysis using MAGMA (*de Leeuw et al., 2015*) implemented in FUMA (*Watanabe et al., 2017*). MAGMA uses input GWAS summary statistics to compute gene-based p-values. For this analysis, the gene-based p-value is computed for protein-coding genes by mapping SNPs to genes if SNPs are located within the genes. We set the gene window 2 kb upstream and 1 kb downstream from the genes to include regulatory elements and analyzed 19,933 genes. We used the default settings and LD information from East Asian ancestry subjects from 1KGP3 as a reference. We set the p-value threshold for the test to $5.0 \times 10^{-8}$ (not a gene-wide threshold).

## eQTL analysis

We obtained transcript data from the Genotype-Tissue Expression (GTEx) v8 (*Consortium, 2015*). We examined eQTL data in all available tissues in GTEx to determine the association between gene expression and the leading SNPs within the genome-wide significant locus. We set the significance threshold for eQTL as an FDR < 0.05.

## SMR

SMR is used to determine associations between genetically determined traits, such as gene expression and protein abundance, and a complex trait of interest, such as OPLL. This analysis is designed to test whether the effect size of an SNP on the phenotype is mediated by gene expression. We used SMR software (*Zhu et al., 2016*). We used OPLL summary statistics data and eQTL data obtained from GTEx v7 (*Consortium, 2015*), which is the same build as in this study (GRCh37). We evaluated heterogeneity in dependent instruments (HEIDI) using multiple SNPs in a *cis*-eQTL region to distinguish pleiotropy from the linkage. As previously reported, we set the threshold for the HEIDI test to 0.05 and the threshold for SMR to $8.4 \times 10^{-6}$ (*Zhu et al., 2016*).

## Partitioning heritability enrichment analysis

We performed stratified LD score regression using 220 cell-type-specific annotations of four histone marks (H3K4me1, H3K4me3, H3K9ac, and H3K27ac) (*Roadmap Epigenomics Consortium et al., 2015*). We divided the 220 cell-type-specific annotations into 10 cell-type groups (10 in adrenal/pancreas, 34 in central nervous system, 15 in cardiovascular, 6 in connective/bone, 44 in gastrointestinal, 67 in immune/hematopoietic, 5 in kidney, 6 in liver, 10 in skeletal muscle, and 23 in other). We assessed heritability enrichment in histone marks of 220 individual cell types and ten cell type groups, as described by *Finucane et al., 2015*. The regression analysis excluded variants within the major histocompatibility complex (MHC) region (chromosome 6: 25–34 Mb). We defined significant heritability enrichment as those with an FDR < 0.05.

## Confirmation of gene expression in ligament tissue

We examined the gene expression in the tissue. We used the deposited RNA-seq data in spinal ligament (yellow ligament) in patients with OPLL and cervical CSM (GSE126060) and human ligament cells after chondrogenic differentiation and control (GSE188759) (*Tachibana et al., 2022*). The presence or absence of gene expression was confirmed by the transcripts per kilobase million (TPM) values in each tissue and cell in both data, followed by a comparison of the expression levels in the two groups by a *t*-test using R software.

## scRNAseq data processing

We used available deposited data from two studies using Achilles tendon cells from an ossification model of the mouse Achilles tendon to investigate the expression levels of candidate causal genes in our study: burn/tenotomy heterotopic ossification model (GSE126060) (*Sorkin et al., 2020*) and Achilles tendon puncture model (GSE188758) (*Tachibana et al., 2022*). These data had already been processed into MTX format, and we conducted subsequent data analyses using the R package Seurat (*Hao et al., 2021*).

Regarding GSE126060 (*Sorkin et al., 2020*), we used data from day 0 and day 7 of the replicate 1–4 samples from the deposited data. First, we filtered out cells with less than 1000 genes per cell and a mitochondrial read content greater than 5%. After normalization using the NormalizeData function and identifying the independently variable 2000 features in each data set, we selected repeatedly variable features across data sets for data integration with the SelectIntegrationFeatures function. We selected anchors using the FindIntegrationAnchors function and used them to integrate the data sets using the IntegrateData function with the default parameter. Next, we scaled this integrated data using the ScaleData function and ran PCA. We then conducted dimensionality reduction with UMAP using the top 20 PCs. After computing k for the k-nearest neighbor algorithm using the top 20 PCs with the FindNeighbors function, we conducted the clustering with the FindClusters function. Next, we annotated the clusters based on the expression of marker genes in the cells comprising the ossified ligament: mesenchymal cell (*Pdgfra*, *Prrx1*, *Clec3b*, and *Dpt*), dendritic cell (*Cd209a* and *Flt3*), endothelial cell (*Emcn*, *Pecam1*, and *Sox18*), lymphocyte (*Ccr7*, *Ms4a4b*, and *Ms4a1*), neuromuscular (*Pax7* and *Ncam1*), pericyte/smooth muscle (*Abcc9*, *Rgs5*, *Acta2*, *Pdgfrb*, and *Des*), macrophage (*Lyz2*, *Cd14*, and *Cd68*), and nerve (*Sox10*, *Plp1*, *Mbp*, and *Mpz*) (*Tachibana et al., 2022*). Finally, we investigated the expression levels of candidate genes found in GWAS meta-analyses, gene-based association analysis, and SMR in each cluster.

In addition, we also conducted an analysis using the data from GSE188758 (*Tachibana et al., 2022*). Since this is a single data set from five mice, we did not conduct data integration. Other than that, we processed the data in the same manner described above and performed clustering and sub-clustering on this data to evaluate the expression of the candidate genes found in our study.

## Genetic correlation

We estimated the genetic correlations using a bivariate LD score regression (*Bulik-Sullivan et al., 2015*) using recently published GWAS results for Japanese: 96 complex traits (*Akiyama et al., 2019*; *Akiyama et al., 2017*; *Ishigaki et al., 2020*; *Kanai et al., 2018*; see *Supplementary file 13* for the traits analyzed). In these reports, GWASs were conducted for 42 diseases (*Ishigaki et al., 2020*), 58 quantitative traits (*Kanai et al., 2018*), BMI (*Akiyama et al., 2017*), and height (*Akiyama et al., 2019*) (total of 102 traits). We could not calculate the following six traits for genetic correlation with OPLL due to the small sample size: biliary tract cancer, endometriosis, hematological malignancy, interstitial lung disease, periodontal disease, and E/A ratio.

We excluded variants found in the MHC region from the analysis because of their complex LD structure. We set the significance threshold for genetic correlations as FDR < 0.05. We evaluated the genetic correlation only for ALL-OPLL because the sample sizes of the C- and T-OPLL groups were too small for this analysis.

## MR

We applied two-sample MR methods that handle summary statistics from separate studies to evaluate the causality of BMI, T2D, cerebral aneurysm, and BMD on OPLL using the R package 'TwoSampleMR' (*Hemani et al., 2018*). Regarding BMI, we reconstructed trans-ancestral meta-analysis data using Japanese and European GWAS results in the same way as previously reported (*Akiyama et al., 2017*; *Locke et al., 2015*). Regarding T2D, cerebral aneurysm, and BMD, we used publicly available results of East Asian meta-analysis of GWASs for T2D (*Spracklen et al., 2020*), mainly European meta-analysis of GWAS for cerebral aneurysm (*Bakker et al., 2020*), and European GWAS for BMD (*Kemp et al., 2017*). In the SNP selection, we extracted the lead SNPs for each study as the instrumental variables. When the lead SNP was not present in the GWAS meta-analysis for OPLL, we selected proxy SNPs highly correlated with the original variants ($r^2 > 0.8$). If there were no proxy SNPs that met the criteria, we excluded the SNPs from the analysis. The details of the instrument variables in each MR are shown in *Figure 4—figure supplement 1* and *Supplementary file 14*.

We conducted additional MR methods in addition to the conventional IVW method for sensitivity analyses: the MR-Egger method (*Bowden et al., 2015*; *Burgess and Thompson, 2017*) and the simple and weighted median methods (*Bowden et al., 2016*). In addition, we conducted subtype-stratified analyses using summary statistics from C- and T-OPLL GWAS and examined the differences between OPLL subtypes. The number of SNPs used in each analysis is listed in *Supplementary file 14*. We also assessed the potential bias in the results of MR with leave-one-out analysis and funnel plots.

We performed reverse-direction MR using the lead SNPs in the significant locus in the meta-analysis for ALL-OPLL as instrumental variables. Parts of these OPLL-associated SNPs were not present within the BMI and cerebral aneurysm data sets, although all were contained in the dataset of T2D and BMD. Therefore, in the analyses for BMI and cerebral aneurysm, we substituted them with the proxy SNP in the same way described above (*Figure 4—figure supplement 1*).

A participant overlap between the samples used to estimate genetic associations with the exposure and the outcome in two-sample MR can bias results (*Burgess et al., 2016*). Therefore, using exposure and outcome instrument variables estimated in non-overlapping samples is preferable. We checked the cohort data used in our MR and found no overlap with the OPLL case sample, although the control samples used in the OPLL study overlapped with up to 2.2% of samples used in the BMI study and 3.4% in the T2D study. According to a simulation study of the association between sample overlap and the degree of bias in instrumental variable analysis, an unbiased estimate is obtained if the overlapping sample includes only control samples for the binary outcome (*Burgess et al., 2016*).

## Additional MR for obesity-related traits

Based on the MR results for the above four traits, we used BMI data only for Japanese (*Akiyama et al., 2017*). We conducted MR to evaluate the causal effect of BMI on OPLL in the same way as above as a sensitivity analysis.

In addition, we conducted MR for obesity-related traits in the same manner as described above. We used the data from the large GWAS meta-analyses for obesity-related traits from UKBB and Giant Consortium: BMI, WHR, and WHRadjBMI (*Pulit et al., 2019*).

## Comparison of the effect sizes of the SNPs between the GWAS meta-analyses for OPLL and BMI

After pruning the SNPs, we evaluated the correlations of the effect sizes of the SNPs between the GWAS meta-analyses for OPLL (ALL-, C-, and T-OPLL) and the GWAS for BMI for sets of SNPs stratified by the thresholds based on the GWAS p-values for each trait. We used the results of the GWAS meta-analysis of OPLL and the Japanese GWAS of BMI (*Akiyama et al., 2017*) for this analysis. First, we extracted SNPs with MAF $\geq$ 0.01, shared between the meta-analyses for OPLL and BMI. Next, we conducted LD pruning of the SNPs for the SNP pairs in LD ($r^2 \geq 0.5$) using 1KGP3 East Asian and JEWEL 3K data by PLINK. Finally, we used 367,672 SNPs in subsequent analyses. We calculated the correlation of the effect sizes of the SNPs between the GWAS meta-analyses for OPLL and BMI GWAS for sets of SNPs stratified by the thresholds based on the GWAS p-values in each trait using R software.

## Generation of PRS of BMI and its application to OPLL GWAS samples

We used PRS to investigate the genetic impact of BMI on OPLL. We constructed the PRS of BMI using a pruning and thresholding method (*Khera et al., 2018*; *Figure 5—figure supplement 1*). In the discovery phase, we generated PRS as the sum of risk alleles weighted by the log odds ratio of association estimated in the Japanese GWAS for BMI (*Akiyama et al., 2017*). We pruned SNPs based on nine different LD thresholds $r^2 = 0.1$–$0.9$ in a 250 kb window using PLINK2.0 and constructed 20 PRS using independent SNPs at p-value thresholds of $5.0 \times 10^{-8} \sim 1$ for each LD threshold. In the validation phase, we determined the best pruning parameter with another Japanese dataset in which genotyping was conducted using the Illumina OmniExpressExome chip. We conducted data quality control in the same manner as in the OPLL GWAS. We used the calculated BMI PRS after standardization (mean = 0, standard deviation = 1). In the test phase, we calculated the Spearman's rho score between BMI and PRS to assess the fit of the models and determined $r^2 = 0.9$ and p-value = 0.6 as the best parameter. We applied BMI PRS for cases and controls in the present OPLL GWAS study.

We measured the association between BMI PRS and OPLL using *logistic regression* with principal components 1–10 as covariates for each OPLL dataset. Then, we meta-analyzed the effect sizes of BMI PRS in the three OPLL data sets using the inverse variance method under a fixed-effect model. We conducted this analysis for ALL-OPLL, C-, and T-OPLL. Furthermore, we applied BMI PRS for C- and T-OPLL cases and compared the effect of OPLL PRS on OPLL subtypes using logistic regression with principal components 1–10 as covariates.

To compare the effect size of BMI PRS in OPLL with T2D, for which BMI is known to be one of the major causes, we performed the same BMI PRS scoring on the T2D data set (39,758 cases and 111,487 controls). Unfortunately, the T2D data set overlaps almost entirely with the data from the discovery phase of the BMI PRS, which carries a high risk of statistical inflation due to model overfitting. However, since no other data was available for T2D, the comparison was made in this manner, knowing the above points.

## Replication analysis for study validation

For replication cases, we extracted DNA from the peripheral blood of 230 OPLL patients, independent of the cases used in the GWAS meta-analysis described above. We genotyped them using Illumina Asian Screening Array. All case samples used here were of the T-OPLL subtype. For replication controls, we used genotyping data for 1889 Japanese individuals from the BBJ project genotyped with Illumina Asian Screening Array. Sample and SNP QC, phasing, and imputation were performed similarly to the GWAS meta-analysis described above. Finally, we conducted a logistic regression analysis using 212 case and 1866 control samples with the top 10 PCs as covariates to confirm the effect size of the lead SNP in each region in the GWAS meta-analysis. To validate the results of our GWAS meta-analysis, we performed a binomial test to assess the concordance in the direction of the betas in the original GWAS meta-analysis and the association analysis with the replication data.

In addition, for the SNPs used as instrumental variables in MR, we updated the SNP statistics by adding replication data to the data in sets 1–3 using meta-analysis with the inverse variance method under a fixed-effect model. Using these updated SNP statistics, we re-conducted MR and evaluated the changes in the results.

## Acknowledgements

We thank the OPLL patients and their families who participated in this study and the OPLL patient network in Japan (Zensekichuren. President, Yasuko Masuda). This work was supported by a JSPS KAKENHI Grant (no. 22H03207 to S Ikegawa and no. 19K09566 to TE), a grant from Japan Orthopaedics and Traumatology Foundation (no. 452 to Y Koike), a grant from AO Spine Japan Research (to M Takahata), a grant from Suhara Memorial Foundation (to M Takahata), grants from Japan Agency for Medical Research and Development (AMED) (no. JP22ek0109555, JP21tm0424220, and JP21ck0106642 to CT), and the JCR Grant for Promoting Basic Rheumatology (to CT).

## Additional information

### Group author details

**Genetic Study Group of Investigation Committee on Ossification of the Spinal Ligaments**
**Tsuji Takashi**: Department of Orthopaedic Surgery, School of Medicine, Keio University, Tokyo, Japan; **Miyamoto Takeshi**: Department of Orthopaedic Surgery, School of Medicine, Keio University, Tokyo, Japan; **Chiba Kazuhiro**: Department of Orthopaedic Surgery, School of Medicine, Keio University, Tokyo, Japan; **Matsumoto Morio**: Department of Orthopaedic Surgery, School of Medicine, Keio University, Tokyo, Japan; **Toyama Yoshiaki**: Department of Orthopaedic Surgery, School of Medicine, Keio University, Tokyo, Japan; **Inose Hiroyuki**: Department of Orthopaedic Surgery, Tokyo Medical and Dental University, Tokyo, Japan; **Yoshii Toshitaka**: Department of Orthopaedic Surgery, Tokyo Medical and Dental University, Tokyo, Japan; **Kawabata Shigenori**: Department of Orthopaedic Surgery, Tokyo Medical and Dental University, Tokyo, Japan; **Okawa Atsushi**: Department of Orthopaedic Surgery, Tokyo Medical and Dental University, Tokyo, Japan; **Yamazaki Masashi**: Department of Orthopaedic Surgery, Faculty of Medicine, University of Tsukuba, Tsukuba, Japan; **Masao Koda**: Department of Orthopaedic Surgery, Faculty of Medicine, University of Tsukuba, Tsukuba, Japan; **Koike Yoshinao**: Department of Orthopedic Surgery, Hokkaido University Graduate School of Medicine, Sapporo, Japan; **Takahata Masahiko**: Department of Orthopedic Surgery, Hokkaido University Graduate School of Medicine, Sapporo, Japan; **Endo Tsutomu**: Department of Orthopedic Surgery, Hokkaido University Graduate School of Medicine, Sapporo, Japan; **Imagama Shiro**: Department of Orthopedics, Nagoya University Graduate School of Medicine, Nagoya, Japan; **Kobayashi Kazuyoshi**:

Department of Orthopedics, Nagoya University Graduate School of Medicine, Nagoya, Japan; **Nakashima Hiroaki**: Department of Orthopedics, Nagoya University Graduate School of Medicine, Nagoya, Japan; **Ando Kei**: Department of Orthopedics, Nagoya University Graduate School of Medicine, Nagoya, Japan; **Kaito Takashi**: Department of Orthopaedic Surgery, Osaka University Graduate School of Medicine, Osaka, Japan; **Kashii Masafumi**: Department of Orthopaedic Surgery, Osaka University Graduate School of Medicine, Osaka, Japan; **Kato Satoshi**: Department of Orthopaedic Surgery, Graduate School of Medical Science, Kanazawa University, Kanazawa, Japan; **Kawaguchi Yoshiharu**: Department of Orthopaedic Surgery, Toyama University, Toyama, Japan; **Sakai Hiroaki**: Department of Orthopaedic Surgery, Spinal Injuries Center, Iizuka, Japan; **Shindo Shigeo**: Department of Orthopedics, Kudanzaka Hospital, Tokyo, Japan; **Taniguchi Yuki**: Department of Orthopaedic Surgery, Faculty of Medicine, The University of Tokyo, Tokyo, Japan; **Takeuchi Kazuhiro**: Department of Orthopaedic Surgery, National Okayama Medical Center, Okayama, Japan; **Maeda Shingo**: Department of Medical Joint Materials, Graduate School of Medical and Dental Sciences, Kagoshima University, Kagoshima, Japan; **Ichiro Kawamura**: Department of Medical Joint Materials, Graduate School of Medical and Dental Sciences, Kagoshima University, Kagoshima, Japan; **Nakajima Hideaki**: Department of Orthopaedics and Rehabilitation Medicine, Faculty of Medical Sciences, University of Fukui, Fukui, Japan; **Baba Hisatoshi**: Department of Orthopaedics and Rehabilitation Medicine, Faculty of Medical Sciences, University of Fukui, Fukui, Japan; **Uchida Kenzo**: Department of Orthopaedics and Rehabilitation Medicine, Faculty of Medical Sciences, University of Fukui, Fukui, Japan; **Mori Kanji**: Department of Orthopaedic Surgery, Shiga University of Medical Science, Otsu, Japan; **Seichi Atsushi**: Department of Orthopedics, Jichi Medical University, Shimotsuke, Japan; **Kimura Atsushi**: Department of Orthopedics, Jichi Medical University, Shimotsuke, Japan; **Fujibayashi Shunsuke**: Department of Orthopaedic Surgery, Graduate School of Medicine, Kyoto University, Kyoto, Japan; **Kanchiku Tsukasa**: Department of Orthopedic Surgery, Yamaguchi University Graduate School of Medicine, Ube, Japan; **Watanabe Kei**: Department of Orthopaedic Surgery, Niigata University Medical and Dental General Hospital, Niigata, Japan; **Tanaka Toshihiro**: Department of Orthopaedic Surgery, Hirosaki University Graduate School of Medicine, Hirosaki, Japan; **Kida Kazunobu**: Department of Orthopaedic Surgery, Kochi Medical School, Nankoku, Japan; **Kobayashi Sho**: Department of Orthopaedic Surgery, Hamamatsu University School of Medicine, Hamamatsu, Japan; **Takahashi Masahito**: Department of Orthopaedic Surgery, Kyorin University School of Medicine, Tokyo, Japan; **Yamada Kei**: Department of Orthopaedic Surgery, Kurume University School of Medicine, Kurume, Japan; **Terao Chikashi**: Laboratory for Statistical and Translational Genetics, Center for Integrative Medical Sciences, RIKEN, Yokohama, Japan; **Ikegawa Shiro**: Laboratory for Bone and Joint Diseases, Center for Integrative Medical Sciences, RIKEN, Tokyo, Japan

**Competing interests**

Yoshiharu Kawaguchi: Consulting fees from Medacta International. Genetic Study Group of Investigation Committee on Ossification of the Spinal Ligaments: Masashi Yamazaki, Atsushi Okawa: representatives of Japanese Organization of the Study for Ossification of Spinal Ligament. The other authors declare that no competing interests exist.

**Funding**

| Funder | Grant reference number | Author |
|---|---|---|
| Japan Society for the Promotion of Science | 22H03207 | Shiro Ikegawa |
| Japan Society for the Promotion of Science | 19K09566 | Tsutomu Endo |
| Japan Orthopaedics and Traumatology Foundation | 452 | Yoshinao Koike |
| AO Spine Japan Research | | Masahiko Takahata |
| Suhara Memorial Foundation | | Masahiko Takahata |

| Funder | Grant reference number | Author |
| --- | --- | --- |
| Japan Agency for Medical Research and Development | JP22ek0109555 | Chikashi Terao |
| Japan Agency for Medical Research and Development | JP21tm0424220 | Chikashi Terao |
| Japan Agency for Medical Research and Development | JP21ck0106642 | Chikashi Terao |
| JCR Grant for Promoting Basic Rheumatology | | Chikashi Terao |

The funders had no role in study design, data collection and interpretation, or the decision to submit the work for publication.

## Author contributions

Yoshinao Koike, Conceptualization, Resources, Formal analysis, Funding acquisition, Visualization, Methodology, Writing - original draft, Project administration, Writing – review and editing; Masahiko Takahata, Conceptualization, Resources, Formal analysis, Funding acquisition, Visualization, Writing - original draft, Project administration, Writing – review and editing; Masahiro Nakajima, Conceptualization, Resources, Formal analysis, Funding acquisition, Writing - original draft, Project administration, Writing – review and editing; Nao Otomo, Formal analysis, Writing - original draft; Hiroyuki Suetsugu, Xiaoxi Liu, Kohei Tomizuka, Keiko Hikino, Formal analysis; Tsutomu Endo, Resources, Formal analysis, Funding acquisition; Shiro Imagama, Resources, Funding acquisition; Kazuyoshi Kobayashi, Takashi Kaito, Satoshi Kato, Yoshiharu Kawaguchi, Masahiro Kanayama, Hiroaki Sakai, Takashi Tsuji, Takeshi Miyamoto, Hiroyuki Inose, Toshitaka Yoshii, Masafumi Kashii, Hiroaki Nakashima, Kei Ando, Yuki Taniguchi, Kazuhiro Takeuchi, Shingo Maeda, Hideaki Nakajima, Kanji Mori, Atsushi Seichi, Shunsuke Fujibayashi, Tsukasa Kanchiku, Kei Watanabe, Toshihiro Tanaka, Kazunobu Kida, Sho Kobayashi, Masahito Takahashi, Kei Yamada, Genetic Study Group of Investigation Committee on Ossification of the Spinal Ligaments, Resources; Shuji Ito, Resources, Formal analysis; Yusuke Iwasaki, Shumpei Niida, Kouichi Ozaki, Yukihide Momozawa, Data curation; Yoichiro Kamatani, Masashi Yamazaki, Atsushi Okawa, Morio Matsumoto, Norimasa Iwasaki, Supervision; Hiroshi Takuwa, Hsing-Fang Lu, Writing - original draft; Chikashi Terao, Conceptualization, Formal analysis, Funding acquisition, Methodology, Project administration, Writing – review and editing; Shiro Ikegawa, Conceptualization, Methodology, Writing – review and editing, Funding acquisition

## Author ORCIDs

Yoshinao Koike (ID) http://orcid.org/0000-0002-5431-5572
Hiroyuki Inose (ID) http://orcid.org/0000-0003-4195-2545
Yuki Taniguchi (ID) http://orcid.org/0000-0002-2329-123X
Sho Kobayashi (ID) http://orcid.org/0000-0002-8002-2001
Shumpei Niida (ID) http://orcid.org/0000-0002-6192-8035
Chikashi Terao (ID) http://orcid.org/0000-0002-6452-4095

## Ethics

Human subjects: All study participants provided informed consent, and the study design was approved by ethical committees at RIKEN Centers for Integrative Medical Sciences (approval ID: 17-16-39), Hokkaido University (approval ID: 16-059), and all other participating institutes.

## Decision letter and Author response

Decision letter https://doi.org/10.7554/eLife.86514.sa1
Author response https://doi.org/10.7554/eLife.86514.sa2

# Additional files

## Supplementary files

• Supplementary file 1. Characteristics of the subjects.

- Supplementary file 2. Independent signals identified by a conditional analysis.
- Supplementary file 3. Functional annotation of SNPs correlated with previously unreported OPLL signals ($r^2 > 0.8$).
- Supplementary file 4. Causal variants estimated by a Bayesian statistical fine-mapping analysis.
- Supplementary file 5. Gene set enrichment analysis.
- Supplementary file 6. Gene-based analysis.
- Supplementary file 7. eQTL analyses for the OPLL-associated SNPs.
- Supplementary file 8. Summary data-based Mendelian randomization.
- Supplementary file 9. Enrichment analysis for active enhancer by cell groups.
- Supplementary file 10. Enrichment analysis for active enhancer by cell types.
- Supplementary file 11. Genome-wide significant loci in cervical/thoracic OPLL.
- Supplementary file 12. Details of genes analyzed by gene expression analysis.
- Supplementary file 13. Genetic correlation between OPLL and other disease or quantitative trait.
- Supplementary file 14. Instrument variables to analyze the causal effect on OPLL.
- Supplementary file 15. MR analyses inferring causality of body mass index, type 2 diabetes, cerebral aneurysm, and bone mineral density on OPLL.
- Supplementary file 16. Estimates of Egger intercept in Mendelian randomization for four traits and OPLL.
- Supplementary file 17. Instrument variables in the reverse-direction Mendelian randomization analysis.
- Supplementary file 18. Reverse-direction Mendelian randomization analysis.
- Supplementary file 19. Results of Mandelian rondomization of body mass index and OPLL using Japanese data only.
- Supplementary file 20. Instrument variables used in additional Mendelian randomization for obesity related traits.
- Supplementary file 21. MR analyses inferring causality of the obesity-related traits on OPLL.
- Supplementary file 22. Estimates of Egger intercept in Mendelian randomization for obesity-related traits and OPLL.
- MDAR checklist

## Data availability

Full GWAS results will be available after acceptance via the website of the Japanese ENcyclopedia of GEnetic associations by Riken (JENGER, http://jenger.riken.jp/en/).

The following dataset was generated:

| Author(s) | Year | Dataset title | Dataset URL | Database and Identifier |
|-----------|------|---------------|-------------|-------------------------|
| Koike et al | 2023 | Case-control GWAS ID 134 | http://jenger.riken.jp/en/result | Japanese ENcyclopedia of GEnetic associations by Riken, 134 |

The following previously published datasets were used:

| Author(s) | Year | Dataset title | Dataset URL | Database and Identifier |
|-----------|------|---------------|-------------|-------------------------|
| Ishigaki et al | 2020 | Case-control GWAS IDs 28-104 (non sex-stratified data) | http://jenger.riken.jp/en/result | Japanese ENcyclopedia of GEnetic associations by Riken, ID 28-104 |

*Continued on next page*

*Continued*

| Author(s) | Year | Dataset title | Dataset URL | Database and Identifier |
|---|---|---|---|---|
| Kanai M, Akiyama M, Takahashi A, Matoba N, Momozawa Y, Ikeda M, Iwata N, Ikegawa S, Hirata M, Matsuda K, Kubo M, Okada Y, Kamatani Y | 2018 | QTL GWAS IDs 7-122 | http://jenger.riken.jp/en/result | Japanese ENcyclopedia of GEnetic associations by Riken, ID 7-122 |
| Akiyama M, Okada Y, Kanai M, Takahashi A, Momozawa Y, Ikeda M, Iwata N, Ikegawa S, Hirata M, Matsuda K, Iwasaki M, Yamaji T, Sawada N, Hachiya T, Tanno K, Shimizu A, Hozawa A, Minegishi N, Tsugane S, Yamamoto M, Kubo M, Kamatani Y | 2017 | QTL GWAS IDs 1-5 | http://jenger.riken.jp/en/result | Japanese ENcyclopedia of GEnetic associations by Riken, ID 1-5 |
| Spracklen et al | 2020 | GWAS summary statistics | https://blog.nus.edu.sg/agen/summary-statistics/t2d-2020 | Asian Genetic Epidemiology Network, T2D_ALL_PRIMARY |
| Bakker MK, Veldink J, Ruigrok Y | 2020 | Intracranial aneurysm genome-wide association study summary statistics 2020 | https://doi.org/10.6084/m9.figshare.11303372 | figshare, 10.6084/m9.figshare.11303372 |
| Kemp et al | 2017 | UKBB eBMD GWAS Data Release 2017 | http://www.gefos.org/?q=content/ukbb-ebmd-gwas-data-release-2017 | GEnetic Factors for OSteoporosis Consortium, ukbb-ebmd-gwas-data-release-2017 |
| Locke et al | 2015 | GWAS Anthropometric 2015 BMI Summary Statistics | http://portals.broadinstitute.org/collaboration/giant/index.php/GIANT_consortium_data_files#GWAS_Anthropometric_2015_BMI_Summary_Statistics | GIANT consortium, GIANT_consortium_data_files#GWAS_Anthropometric_2015_BMI_Summary_Statistics |
| Pulit SL | 2018 | Summary-level data from meta-analysis of fat distribution phenotypes in UK Biobank and GIANT (Meta-analysis of bmi) | https://doi.org/10.5281/zenodo.1251813 | Zenodo, 10.5281/zenodo.1251813 |
| Menon R, Levi B, Huber A, Sorkin M, Marini S | 2019 | Role of myeloid cells in heterotopic ossification (HO) in a burn and incision-induced mouse model | https://www.ncbi.nlm.nih.gov/geo/query/acc.cgi?acc=GSE126060 | NCBI Gene Expression Omnibus, GSE126060 |
| Tachibana N, Saito T | 2022 | scRNA-seq of Achilles tendons from Prg4-CreERT2;R26-tdTomato ATP mice | https://www.ncbi.nlm.nih.gov/geo/query/acc.cgi?acc=GSE188758 | NCBI Gene Expression Omnibus, GSE188758 |
| Tachibana N, Saito T | 2022 | RNA-seq analysis of human spinal ligaments | https://www.ncbi.nlm.nih.gov/geo/query/acc.cgi?acc=GSE188760 | NCBI Gene Expression Omnibus, GSE188760 |
| Tachibana N, Saito T | 2022 | RNA-seq in chondrocyte differentiation of human spinal ligament cells | https://www.ncbi.nlm.nih.gov/geo/query/acc.cgi?acc=GSE188759 | NCBI Gene Expression Omnibus, GSE188759 |

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

## Appendix 1

### SNP–obesity interaction

We evaluated the SNP–obesity interaction during the development of OPLL. We first calculated the best-guess genotypes for lead SNPs in 14 significant loci of the OPLL GWAS meta-analysis (ALL-OPLL) using PLINK. We then scored each individual based on the best-guess genotypes of each of the 14 SNPs: risk allele/risk allele = 2, risk allele/non-risk allele = 1, non-risk allele/non-risk allele = 0. Furthermore, we scored each individual according to the WHO classification of obesity: BMI < 18.5 (underweight) scored 0; $18.5 \leq BMI < 25.0$ (normal range) scored 1; $25.0 \leq BMI < 30.0$ (overweight) scored 2; $30.0 \leq BMI < 35.0$ (obese class 1) scored 3; $35.0 \leq BMI < 40.0$ (obese class 2) scored 4; and $40.0 \leq BMI$ (Obese class 3) scored 5 (*World Health Organization, 2000*). We measured the association between the interaction term (SNP*obesity score) and OPLL using *logistic regression* with principal components 1–10 obtained by principal component analysis with SmartPCA (*Patterson et al., 2006*) as covariates.

We found no significant association between the development of OPLL and SNP-obesity interaction.

## Appendix 2

### Comparison of SNP effect sizes between OPLL and ankylosing spondylitis GWASs

Ankylosing spondylitis (AS) is a chronic immune-mediated arthritis characterized by ossified lesions of the spine known as bamboo spine. Although AS is considered a fundamentally different disease from OPLL, some authors reported that some AS patients also had OPLL: 3.5% of Koreans and 15.5% of Mexicans (*Kim et al., 2007*; *Ramos-Remus et al., 1998*).

We used the results of the largest GWAS for AS, which was a study on Europeans (*Ellinghaus et al., 2016*). We compared effect sizes between OPLL and AS GWASs for AS-associated SNPs and found no correlation (*Figure 1—figure supplement 17*). We could not make the same comparison for OPLL-associated SNPs because AS GWAS summary statistics were unavailable.

