## [Editor Report]

This study builds on previous work to explore the genetic causes of ossification of the posterior longitudinal ligament of the spine (OPLL). A meta-Genome wide association study is conducted to increase detection power and a disease subtype analysis is completed that provides new information on if all sites of OPLL have uniform causes. Using additional open-source data, the GWAS results are explored further to find putatively causative genes and to explore causative co-existing conditions such as obesity for OPLL. Overall this study is by far the most complete genetic exploration of this disease to date and is instructive for future studies that will lead to treatments for this condition.

---

## [Decision Letter]

**Decision letter after peer review:**

[Editors’ note: the authors submitted for reconsideration following the decision after peer review. What follows is the decision letter after the first round of review.]

Thank you for submitting the paper “Genetic insights into ossification of the posterior longitudinal ligament of the spine” for consideration by *eLife*. Your article has been reviewed by 3 peer reviewers, and the evaluation has been overseen by a Reviewing Editor and a Senior Editor. The following individual involved in the review of your submission has agreed to reveal their identity: Emma Duncan (Reviewer #3).

Comments to the Authors:

We are sorry to say that, after consultation with the reviewers, we have decided that this work will not be considered further for publication by *eLife*.

A spirited conversation was had among the reviewers regarding this paper. The reviewers unanimously agree that the topic of the paper is exciting and that this body of work has promise. Unfortunately, the reviewers ultimately agreed that this work is not yet ready to be accepted by *eLife*. There are several major issues that are too larger to consider this work for revision alone. The first issue is that this work lacks validation in any way. It is appreciated that a validation cohort may well be impossible at this time, but without other robust validation approaches, this work is too premature for publication. The other issue is with how the results were interpreted. The reviewers recognized that the perfect transcriptomic datasets for use with this work simply do not exist and that the authors cannot use Japanese GWAS data for every trait in their analyses. It was the overall impression that there was a lot of mix-and-matching of datasets going on to suit pre-ordained hypotheses rather than being more hypothesis-free in their co-heritability analyses. This biased/pre-conceived conclusion approach very much taints the discussion of this paper. A true exploration in a hypothesis-free manner is doable and would result in an outstanding paper. This point is the major reason that this work was rejected. We would like to reconsider this paper as a new submission to *eLife* after the authors have carefully explored their data to ensure that the study was conducted rigorously in a bias-free manner.

*Reviewer #1 (Recommendations for the authors):*

Overall, this study uses a comprehensive suite of statistical tools to move from GWAS through to attempts to identify the mechanism.

Strengths of this work include the thoroughness of the in silico genetics investigation of an understudied genetic condition. This study highlights putative causative mechanisms by which OPLL could arise.

There are two main weaknesses of this study. The first is that this study is largely lacking replication. The second issue is that this is all analytical, and there is no functional follow-up or biological validation. As a result, the mechanism is not yet proven.

1. The abstract does not follow the flow of the paper.

2. Lines 191-200: more is needed to justify the statement that “we found several candidate causal genes reported to be related to osteogenesis….” First of all, the genes listed are the closest to the peak genes and so the word causal should be removed. Second, only two of the genes are described in any detail to support this statement.

3. Additional detail is required for the methods for the application of the FUMA package. As presented, this work assumes a substantial understanding of how FUMA works and what the results are, which is not likely to be true for a non-genetics journal such as this one.

4. A similar issue applies as above for the application of SMR.

5. Lines 226-239 – only two of the three genes are mentioned. If nothing is known about the third, this should be stated.

6. Lines 340 – 345: High-fat diet is not considered to increase bone generally in animal models. This section is a misleading selection of references and is not presented in an unbiased fashion. For example, all of the following references found that a high-fat diet was detrimental to bone: (see following Pubmed ID’s, these are just the 2022 reference 19750487, 35788787, 35762155, 35718325, 35578981, 35257177) in animals models. In humans, the issue is more complex. This section needs to be expanded and be more transparent given that the BMI and OPLL link is a central conclusion.

7. Overall, this study is lacking a summarizing conclusion section.

8. The limitations of this work need to be more completely explored.

*Reviewer #2 (Recommendations for the authors):*

Ectopic ossification is a fairly common clinical finding and the posterior longitudinal spinal ligament is a common site for this to occur (hereafter abbreviated as OPLL). Previous work has identified associations between OPLL and various chronic conditions, including positive association with obesity and negative association with osteoporosis. The authors used a variety of genetic approaches to gain insight into OPLL pathogenesis, which is poorly understood. The authors report several genetic associations from GWAS and candidate gene-based associations and infer potential causal relationships from Mendelian randomization. These findings were used to develop a polygenic risk score. They identify H3K27 acetylation in hypertrophic chondrocytes as a potential critical process in OPLL pathogenesis. The single most compelling element of this work was the Mendelian randomization of BMI with thoracic, but not cervical, OPLL. This was the statistically most robust finding and it also resonates intuitively with the idea that a BMI-based association would be demonstrated in the anatomical region more affected by differences in weight-bearing. Positive associations with type 2 diabetes, uterine fibroids, and cataracts as well as negative associations with BMD, cerebral aneurysms, and HDL cholesterol were also noted. This work opens up the field to new experimental work and represents a clear advance in the field.

Strengths: Undertaking a variety of complementary genetic analyses is a clear strength of this work. In addition to the multiple approaches, there were several additional strengths worth noting. The team did an excellent job of exploiting informatics resources to support their investigation. The GWAS meta-analysis was conducted using a large cohort of Japanese subjects, limiting the risk that population stratification drove positive results. The Mendelian randomization used multiple approaches to seek violation of the pleiotropy assumption. They followed a best practice in conducting the “leave one out” analysis. These investigators did the opposite of “turn the crank” labs that have established pipelines for conducting a specific type of analysis and apply it to every problem. Instead, they used multiple tools to focus on a poorly understood disorder.

Weaknesses: The main limitation of this work was the lack of direct experimental evidence to support the role of the identified candidate genes in OPLL pathogenesis.

Validity and Impact: The authors report 14 genetic associations from GWAS, 5 from candidate gene-based associations, and infer 8 potential causal relationships from Mendelian randomization. Additional Mendelian randomization analyses yielded negative results. These findings were used to develop a polygenic risk score. They identify H3K27 acetylation in hypertrophic chondrocytes as a potential critical process in OPLL pathogenesis. The single most compelling element of this work, in my opinion, was the Mendelian randomization of BMI with thoracic, but not cervical, OPLL. This was the statistically most robust finding and it also resonates intuitively with the idea that a BMI-based association would be demonstrated in the anatomical region more affected by differences in weight-bearing. Positive associations with type 2 diabetes, uterine fibroids, and cataracts as well as negative associations with BMD, cerebral aneurysms, and HDL cholesterol were also noted. This work opens up the field to new experimental work and represents a clear advance in the field.

The main points are covered in the public review. In addition, please consider the following:

1. BMI is a fairly sloppy phenotype. It does not really reflect adiposity (eg some highly trained athletes have high BMIs but very low body fat %). Did you either adjust for stature or use another measure of obesity, and if so, what were the results?

2. It is notable that some of the BMD associations you observed were with pediatric rather than adult phenotypes. Do you think that this might reflect dysregulated prolongation of processes that normally are temporally constrained?

3. Did the analysis of genetic correlation and other conditions include any of the following: serum Ca, serum phosphate, aortic stenosis, injury-related ectopic ossification, or renal function? It might be worth listing all the tested conditions in a supplementary table.

4. I do not understand the distinction between diseases and quantitative traits in figure 3. Were the “diseases” coded categorically while the quantitative traits were classified continuously? You list “osteoporosis” as the disease, but most analyses are done on BMD (quantitative trait) or fracture. I suggest merging the 2 parts of the figure and omitting any attempt to distinguish diseases from traits. I like the way this figure displays effect size and statistical robustness together.

*Reviewer #3 (Recommendations for the authors):*

Here the authors have conducted a comprehensive genetic interrogation of the condition OPLL. The authors have shown that this disease is heritable and identified some interesting genome-wide significant loci – although these explain only a small proportion of the overall heritability. Their functional genomics analyses add to the paper, and their results are plausible – in that some of the identified genes are involved in bone metabolism (although they also show a relationship with immune/haematopoietic cell grouping, which is not explored or discussed). Further, that there is support for a genetic relationship between BMI and OPLL in the Japanese population which corresponds to the clinical epidemiology.

Through no fault of their own, some of the authors’ ability to perform and extrapolate gene expression analyses is hampered by a lack of transcriptomic datasets from arguably the most relevant tissue types (including bone, cartilage, ligamentous tissue) within the available datasets worldwide. Similarly, although the authors aimed in their MR analysis to use Japanese GWAS data, they were often constrained to needing to use other ethnic groups (e.g., for the BMD analysis) and thus the MR analyses are not uniform.

The authors have also restricted some of their analyses to 'the usual suspects' (e.g., BMI, T2DM, BMD) and not considered other ossification processes such as ankylosing spondylitis and heterotopic ossification. This seems to be unnecessarily limiting and counterproductive when considering the benefits of hypothesis-free approaches in generating new insights into aetiopathogenesis and future therapeutic directions.

The discussion needs more nuance and finesse, including placing results in context and considering strengths and weaknesses more carefully. Additionally, the authors have focused on new loci throughout this manuscript: in considering overall pathogenesis/aetiology the authors should include known loci in the discussion. The authors are encouraged to think about their results outside a bone filter.

Summary to Authors

1. There are some methodological issues that the authors cannot solve (e.g., lack of transcriptomic sets from all relevant tissue types; lack of GWAS in Japanese populations for some of the MR analyses); these are iterated in the methods and results but should be highlighted as part of the limitations in the discussion.

2. Most results appear robust (apart from T-OPLL data). Some additional analyses (specifically, some further co-heritability analyses) are suggested that would strengthen this manuscript and should be easily do-able within their dataset. The discussion needs some redrafting according to the comments below.

Introduction and methods

1. Why is the epidemiology of OPLL in Japanese individuals so different from other ethnicities? A three-fold difference in a polygenic disease between ethnicities is a massive difference. Although there may not be an answer here, this massive epidemiological difference needs more emphasis.

2. Are there any previous papers assessing heritability for OPLL? Some data regarding genetic epidemiology would add to this paper. Is there any evidence, for example, of a major gene effect on a polygenic background that might explain the ethnicity differences?

3. Three small GWAS, all conducted in a Japanese population, have been meta-analysed. What is the power of each GWAS individually and for meta-analysis overall? There is a large and differential dropout at the QC level (controls >> cases – e.g. set 1 – 92% of cases passed, 85% of controls; set 3: 97% of cases vs. 86% of controls) – Supplementary figure 1. The dropout Is unusually high and the reason for such differing rates is unclear, although the reviewer notes (lines 386-396 and Supplementary Table 1) that cases and controls were genotyped separately and in different laboratories (here the reviewer notes the high number of principal components used in the analysis – to ten). The authors should include this issue in their limitations. What where the co-variates used in the GWAS (?BMI ?weight ?presence/absence of T2DM) should be detailed. An Rsq>0.3 for imputation is generous; a more stringent threshold would be 0.8. could the authors justify this choice?

4. Are there any other GWAS in other ethnicities? [Perhaps not, given how specific this disease appears to be to Japan, and if this is the case it should be mentioned in the discussion. However, if so, this would make the meta-analysis more informative].

5. Figure 1: Manhattan plot for the meta-analysis. Here the authors appear to have somewhat selectively chosen to annotate some of their significant loci and not others. Line 448 suggests that this is because these are the novel loci. The reviewer thinks it would be more appropriate for all genome-wide significant peaks to be similarly annotated (perhaps in different colours for new vs. known loci), to give the reader the full view of the genetic information within this one image rather than having to refer to previous publications.

6. The reviewer notes that the criteria for cases vary between sets 1 and 2 vs. set 3. The authors do not really justify their decision to split the data into cervical and thoracic subsets from an epidemiological (genetic or otherwise) or clinical basis. Are these really discrete clinical entities, with separate epidemiology and disease course? The retro-assessment of C-OPLL and T-OPLL is not detailed (one reviewer? More than one reviewer? Quality control for calling this clinical phenotype?).

7. Supplementary Figure 3 – the locus zoom plots do not show which SNPs are imputed and which are genotyped. This should be added to the diagrams.

8. Line 461 Decisions around the associated locus of a lead variant is usually set by LD rather than physical distance.

9. Line 488 – Gene set enrichment analysis – although this is a reasonable process, either here or elsewhere the authors should provide reference justification for choosing to assess genes based on physical location (within 1Mb of leading SNP plus up to five genes 'closest' to leading SNPs) as there are now notable examples in the literature where the leading SNP signal does not correlate physically to the causative gene, with obesity being a lead example here.

10. Line 498 – eQTL analysis in 'all available tissues' – the authors need to comment in their discussion on the relevance/appropriateness of the tissues captured in GTEx v8 (noting here bone and/or cartilage and/or ligamentous tissue are not included in GTEx v8).

11. Line 503 – unclear why v7 is used here whereas v8 is used above.

12. Line 518ff Genetic correlations were assessed using bivariate LD score regression and Japanese data – it is implied here that this only used Japanese data (which is appropriate) but this should be definitively confirmed.

13. Lines 528ff for the MR study, the authors used GWAS data from varying ethnic backgrounds, including Japanese and European and east-Asian (presumably Chinese+Japanese here), which varied from trait to trait (e.g European GWAS was used for the BMD analysis, and east Asian for the T2DM analysis). This limits the robustness of this section, as these were not equally valid tools for the MR analysis. In particular, the BMI data were informed from both Japanese and European GWAS data. As the relationship between BMI and OPLL is one of the major findings of this paper, this analysis should be repeated using data ONLY from Japanese BMI GWAS, as a sensitivity analysis.

14. Line 555ff – the authors appropriately assessed their data to ensure no overlap between exposure and outcome datasets – but it is not clear what they then did, having found that there were small overlaps.

15. Considering the PRS of BMI wrt application to OPLL GWAS – the authors appropriately use a Japanese-population GWAS here.

16. Line 577 – the reviewer notes in line 582 that having tried out differing r2 and p values they eventually chose r2 of 0.9 tho' P-value=0.6 – this would mean almost any result in the BMI GWAS could be included. From the image Supplementary Figure 20 they could have chosen p=0.05 without much difference in their rho value, and this would seem at least a little more stringent.

Results

1. Line 175FF: have the authors corrected their p-value for significance by this (modest) genomic inflation factor (1.11)?

2. Line 209ff: the authors should make an explicit comment on their lack of enrichment in DXA-assessed BMD in adults (e.g. using data from GEFOS2).

3. Line 219ff: the authors' decision as to which are and which are not plausible causative genes seems arbitrary [based mainly on previous data for monogenic skeletal dysplasias it would seem]. If only these genes for which there is prior knowledge are considered it somewhat defeats the point of trying to gain new knowledge from hypothesis-free approaches. This continues to be an issue in the discussion too. The authors should describe the known function of other genes too.

4. Similarly Line 241ff. Whilst reporting their significant enrichment in the connective/bone cell group, the authors have not mentioned here that they also had significant enrichment for the immune/hematopoietic cell group, with a significant result for skeletal muscle which did not survive after FDR adjustment. These other cell groups are highly relevant to the phenotype and should be mentioned here similarly.

5. The authors' choice for co-heritability studies is limited. The authors have not considered GWAS for AS, for example, which may be highly relevant. The authors should consider an unbiased coheritability analysis rather than only the 'usual suspects' – as the latter will only really confirm prior thoughts and not illustrate novel pathways/mechanisms (for example see the recent paper on IBS).

6. Supp Figure 5: the calling of eight new loci for T-OPLL is over-enthusiastic. There is very little genetic support for ZBTB40, for example (see Locuszoom plot in Supplementary Figure 8), and it's not clear whether this is a genotyped or imputed SNP (not shown in Supplementary table 11). Similar comments are made about the alleged novel loci in plots c, e, f, g, and h. The only believable putative novel loci are shown in plots b and d. The authors should tone down their claims about discrete novel loci for T-OPLL substantially, therefore, throughout the manuscript (including abstract, results, and discussion). Related to this, the authors should reconsider rewriting sections where they feature T-OPLL. In discussing these possible site-specific T-OPLL GWAS results, the authors do not avail themselves of the opportunity to discuss the two plausible loci but only discuss ZBTB40. What is known about genes within the two more plausible loci?

7. Line 265ff the authors should flag at first mention here and in the methods (at line 520) that there is a supplementary table listing the complex traits considered here, especially as this is limited (appropriately) to previous genetic studies in Japanese populations (as per the methods). Although the authors' approach here is appropriate (i.e., limited to the Japanese population), the point is that these GWAS represent only a subset of the many traits mapped by GWAS internationally [this is given later in the paragraph but should be flagged earlier for the reader].

8. Line 271: ossification and bone mineral density accrual are quite different processes. In particular, BMD is likely to be artefactually elevated by the presence of OPLL, if this affects lower spinal areas, as it is also in AS, DISH, etc. The authors need to provide more clarity, nuance, and detail here.

9. The results would benefit from editorial comments being removed from the results and placed in the discussion (e.g. lines 269-271).

10. Line 316ff PRS scores – these β values may be significant but they are small. Can the authors contextualize their results with, say, the PRS for BMI and T2D?

11. The authors have attempted to split their data by clinical phenotype (cervical vs. thoracic OPLL); the data supporting multiple novel loci for T-OPLL do not appear robust from the locuszoom plots and the authors are encouraged to tone down this component substantially (for example, focusing on ZBTB40 as a locus for T-OPLL appears implausible from the Locuszoom plot (one is left with an impression that there was some prior hypothesis contribution to this point [e.g., it being a known BMD locus]) whereas other much more plausible loci (based on the Locuszoom images) do not receive similar attention).

Discussion

Their discussion is inadequate; and needs much more care and non-biased interpretation, and detailing of the [many] limitations. Also, there is an obvious clinical question, that they haven't posed: has weight loss been trialled as a therapeutic approach to mxmt of OPLL?

1. The authors' functional assessment in their first paragraph on bone formation needs toning down. Certainly, the authors' results are enriched for signals in bone metabolism pathways – but they don't know that these are necessarily anabolic, or anti-catabolic; and line 324ff bone formation and resorption are normally tightly coupled processes so it would be unusual for these to be uncoupled in OPLL uniquely. Moreover, the authors have mainly featured those genes that suit their argument; and have not commented, for example, that their gene enrichment analyses also identified immune/haematopoietic cell groups. Allowing for the time being their comment to stand: is there any evidence that bisphosphonates (that suppress both bone formation and bone resorption) improve outcomes with OPLL?

2. The relationship between EB and osteoporosis needs far more nuance if the authors are to include it in their argument regarding PLEC: it is thought to be multifactorial and related more to nutrition, drugs, mobility, etc, and there is not, to my knowledge, a known underlying mechanistic link directly related to the mutation and bone per se.

3. The benefits of considering this approach as novel knowledge-generating have been somewhat lost by the authors only focusing on bone in their discussion, and as they mostly do not mention information about previously reported loci it's hard to contextualise their newer findings with older knowledge regarding bone pathways vs. other pathways in terms of OPLL overall.

4. Lines 340ff – here the authors should be careful not to conflate BMD with ossification processes, and this section requires more nuance. The relationship between obesity and BMD is extremely contentious, and only referring to Lv et al. is a very simplistic reflection of the literature. (most of this paragraph really discusses the relationship between BMI and BMD, and not, as is pertinent here, BMD and OPLL, so it should be revised to focus on the newer loci presented here).

5. The authors have missed an opportunity to consider a relationship between OPLL and other ossifying processes, such as ankylosing spondylitis and heterotopic ossification. Whilst these may not have had GWAS performed in Japanese populations the authors have not always had Japanese population data for those studies that they have included (e.g., BMD).

6. The PRS suggests that there is a genetic relationship between BMI and OPLL, with BMI driving OPLL. However, the β suggests the clinical impact is extremely small. Contextualising their results with other PRS would help the reader understand the clinical impact of their findings.

7. The authors do not discuss the strengths and weaknesses of their paper, which would add sobriety to their discussion.

8. Paradoxically, the authors focus a lot on talking about novel treatments, etc but by only looking at previously known pathways they've limited the ability to look at these. Also, the most obvious way to manage high BMI would be weight loss. Not sure this really represents a novel treatment for OPLL, as such.

9. The authors have also not contextualized their results in light of previous data. For example, here they have shown heritability of OPLL of >50%. Have there been any previous studies of heritability?

---

## [Author Response]

Editors’ note: the authors resubmitted a revised version of the paper for consideration. What follows is the authors’ response to the first round of review.

Reviewer #1 (Recommendations for the authors):Overall, this study uses a comprehensive suite of statistical tools to move from GWAS through to attempts to identify the mechanism.Strengths of this work include the thoroughness of the in silico genetics investigation of an understudied genetic condition. This study highlights putative causative mechanisms by which OPLL could arise.There are two main weaknesses of this study. The first is that this study is largely lacking replication. The second issue is that this is all analytical, and there is no functional follow-up or biological validation. As a result, the mechanism is not yet proven.

We thank you for the comprehensive review of our manuscript and valuable suggestions. We have again discussed the two weaknesses you have pointed out.

As for the lack of replication, we added new replication data and evaluated the reproducibility. Our study originally conducted meta-analyses using three datasets. The result showed that the directions of effect sizes of all 14 lead SNPs in the significant loci were consistent across the three cohorts. The analyses of OPLL subtypes also showed that the directions of effect sizes for most lead SNPs were consistent among the three cohorts. With these results, we believed that our results were valid. However, as you pointed out, we understood that to make our report more reliable, it is important to further check the reproducibility using another data set even after confirming that the directions of the signals in the meta-analysis are consistent in each cohort. Therefore, we evaluated the reproducibility of the lead SNP signals using OPLL data independent of the data used in the meta-analysis. As for ALL-OPLL, we confirmed that the direction of betas for the lead SNPs in most regions (13/14) is consistent between the original GWAS meta-analysis and the replication study (P value of the binomial test was 1.83E-03). In this way, we were then able to further reinforce the validity of our data.

Furthermore, to evaluate the validity of the results of Mendelian randomization (MR), which is one of the main analyses in this study to assess the causal relationship between other traits and OPLL, we re-conducted MR with the addition of data from the replication cohort and compared them with the original data. As a result, the MR conducted again successfully reinforced the validity of our original results.

As for functional follow-up or biological validation, we evaluated the expression levels of the candidate genes discovered in our study using gene expression data from spinal ligament tissue and related tissues/cells. As you pointed out, biological validation, such as confirmation of gene expression in the actual target tissue, is very important to examine the involvement of the target gene in the disease. Although gene expression data for spinal ligament tissue is scarce, there were some useful available deposited data in a recently published paper (PMID: 35984875): gene expression data in a spinal ligament in patients with OPLL and cervical spondylotic myelopathy (CSM) (GSE126060), and in human ligament cells following chondrogenic differentiation (GSE126060). We were able to confirm the expression of the candidate genes in these tissues and cells, reinforcing the validity of this study. To further validate our findings, we evaluated the expression of the candidate genes using available single-cell RNA sequencing data from a mouse Achilles tendon ossification model (GSE126060 and GSE188758). As a result, we could confirm the gene expression with single-cell resolution.

By improving these two points you mentioned, we believe our study's findings have become more robust. We would like to thank you again for the valuable review comments.

1. The abstract does not follow the flow of the paper.

Thank you for the comments. We have changed the style of the abstract to match the flow of the results.

2. Lines 191-200: more is needed to justify the statement that "we found several candidate causal genes reported to be related to osteogenesis…." First of all, the genes listed are the closest to the peak genes and so the word causal should be removed. Second, only two of the genes are described in any detail to support this statement.

Thank you very much for the comments. We agree that the word “causal” is somewhat overstated for genes closest to lead variants (while closest genes are the best candidates of causal genes in many GWAS papers). We have modified to “potentially causal”.

3. Additional detail is required for the methods for the application of the FUMA package. As presented, this work assumes a substantial understanding of how FUMA works and what the results are, which is not likely to be true for a non-genetics journal such as this one.

Thank you very much for the comments. We agree to provide a more detailed explanation of the FUMA to broad readers. We have added a description of FUMA and the gene-based analysis called MAGMA used in FUMA as follows.

p28, L663

“FUMA is a web-based platform in which we can perform GWAS-related analyses such as gene-based analysis and gene set enrichment analysis (Watanabe et al., 2017).”

p28, L671

“MAGMA uses input GWAS summary statistics to compute gene-based P-values. For this analysis, the gene-based P-value is computed for protein-coding genes by mapping SNPs to genes if SNPs are located within the genes.”

4. A similar issue applies as above for the application of SMR.

Thank you very much for the comments. As you suggested, we also added an explanation for SMR as follows.

p29, L684

“SMR is used to determine associations between genetically determined traits, such as gene expression and protein abundance, and a complex trait of interest, such as OPLL. This analysis is designed to test if the effect size of an SNP on the phenotype is mediated by gene expression.”

5. Lines 226-239 – only two of the three genes are mentioned. If nothing is known about the third, this should be stated.

Thank you very much for the comments. We agree to mention all three genes we found in the SMR. I searched the literature regarding the gene *RP11-967K21.1*, but could not find anything related to OPLL. The following was added to the text as a description of *RP11-967K21.1.*

p11, L251

“As for *RP11-967K21.1*, its function in OPLL development is currently unknown and is expected to be elucidated in future studies.”

6. Lines 340 – 345: High-fat diet is not considered to increase bone generally in animal models. This section is a misleading selection of references and is not presented in an unbiased fashion. For example, all of the following references found that a high-fat diet was detrimental to bone: (see following Pubmed ID's, these are just the 2022 reference 19750487, 35788787, 35762155, 35718325, 35578981, 35257177) in animals models. In humans, the issue is more complex. This section needs to be expanded and be more transparent given that the BMI and OPLL link is a central conclusion.

Thank you very much for the very important remarks. We agree that the selection of the references in the original submission might be misleading and might be regarded as biased.

Many papers in clinical studies show that obesity affects the direction of bone gain, but in animal models, there are two sides to the argument. Rather, high-fat diets are often reported to have a negative effect on bone in animal models. We agree to mention these points fairly. Thus, we modified the manuscript as follows.

p19, L452

“Yamamoto *et al.* reported a high incidence of OPLL in the Zucker fatty rat, a model mouse of obesity with a loss-of-function mutation in the leptin receptor gene (Yamamoto et al., 2004). However, basic studies with small animals have reported conflicting results regarding the effects of obesity on bone formation. In general, high-fat diet (HFD)-induced obesity is detrimental to bone formation and results in low bone density in mice (Zhang et al., 2022). On the other hand, Lv et al. reported that the mRNA level of *Runx2* in bone marrow-derived mesenchymal stem cells from HFD mice was significantly higher; in contrast, that of *Pparγ*, which suppresses osteoblast differentiation and promotes osteoclast differentiation, was significantly lower than the control mice (Lv et al., 2010). Thus, the results of basic experiments are varied and controversial.”

7. Overall, this study is lacking a summarizing conclusion section.

Thank you for the comments. We agree to include a conclusion section; we have added one in the last paragraph of the Discussion section as follows.

p23, L542

In conclusion, this study identified candidate genes in genomic loci associated with OPLL, and subsequent post-GWAS analyses showed a causal relationship between other traits and OPLL: obesity and high BMD. This study will serve as a basis for future research to elucidate the pathogenesis of OPLL in more detail and to develop new treatment methods.

8. The limitations of this work need to be more completely explored.

Thank you very much for your comments. Another reviewer also commented on describing the limitations of this study. Thus, we describe the limitations of this study in the Discussion section as follows.

p22, L525

“There are some limitations in this study. The first is the GWAS sample size. Although this study is the largest OPLL GWAS in the world and used samples collected from facilities throughout Japan, additional samples are needed to clarify the pathogenesis of OPLL further and better define subtype differences. In addition, OPLL GWAS in other ethnic groups has not been reported. We expect future international meta-analysis using data from non-Japanese ethnic groups to elucidate regional differences in the frequencies of OPLL. Second, gene expression data in spinal ligament tissue is scarce. Therefore, we were forced to use tissue data different from ligament tissue, such as GETx (GTEx Consortium, 2015), which did not include bone, cartilage, or ligamentous tissue, and data from a mouse Achilles tendon ossification model, which resembles spinal ligament ossification. We also used bulk data from spinal ligaments, but the sample sizes were limited. It is desirable to increase the number of spinal ligament tissue samples and perform expression analysis using scRNAseq in the future. If so, genes other than those focused on in this study could be identified as causal genes. Another limitation is that the Japanese GWAS data used for genetic correlation analysis were limited to 96 traits, and other traits not analyzed for genetic correlation may be associated with OPLL. In addition, some of the data used for MR were not East Asian. While we cannot immediately improve upon the limitations listed above, we intend to strengthen these areas for future research for OPLL.”

Reviewer #2 (Recommendations for the authors):Ectopic ossification is a fairly common clinical finding and the posterior longitudinal spinal ligament is a common site for this to occur (hereafter abbreviated as OPLL). Previous work has identified associations between OPLL and various chronic conditions, including positive association with obesity and negative association with osteoporosis. The authors used a variety of genetic approaches to gain insight into OPLL pathogenesis, which is poorly understood. The authors report several genetic associations from GWAS and candidate gene-based associations and infer potential causal relationships from Mendelian randomization. These findings were used to develop a polygenic risk score. They identify H3K27 acetylation in hypertrophic chondrocytes as a potential critical process in OPLL pathogenesis. The single most compelling element of this work was the Mendelian randomization of BMI with thoracic, but not cervical, OPLL. This was the statistically most robust finding and it also resonates intuitively with the idea that a BMI-based association would be demonstrated in the anatomical region more affected by differences in weight-bearing. Positive associations with type 2 diabetes, uterine fibroids, and cataracts as well as negative associations with BMD, cerebral aneurysms, and HDL cholesterol were also noted. This work opens up the field to new experimental work and represents a clear advance in the field.Strengths: Undertaking a variety of complementary genetic analyses is a clear strength of this work. In addition to the multiple approaches, there were several additional strengths worth noting. The team did an excellent job of exploiting informatics resources to support their investigation. The GWAS meta-analysis was conducted using a large cohort of Japanese subjects, limiting the risk that population stratification drove positive results. The Mendelian randomization used multiple approaches to seek violation of the pleiotropy assumption. They followed a best practice in conducting the "leave one out" analysis. These investigators did the opposite of "turn the crank" labs that have established pipelines for conducting a specific type of analysis and apply it to every problem. Instead, they used multiple tools to focus on a poorly understood disorder.Weaknesses: The main limitation of this work was the lack of direct experimental evidence to support the role of the identified candidate genes in OPLL pathogenesis.Validity and Impact: The authors report 14 genetic associations from GWAS, 5 from candidate gene-based associations, and infer 8 potential causal relationships from Mendelian randomization. Additional Mendelian randomization analyses yielded negative results. These findings were used to develop a polygenic risk score. They identify H3K27 acetylation in hypertrophic chondrocytes as a potential critical process in OPLL pathogenesis. The single most compelling element of this work, in my opinion, was the Mendelian randomization of BMI with thoracic, but not cervical, OPLL. This was the statistically most robust finding and it also resonates intuitively with the idea that a BMI-based association would be demonstrated in the anatomical region more affected by differences in weight-bearing. Positive associations with type 2 diabetes, uterine fibroids, and cataracts as well as negative associations with BMD, cerebral aneurysms, and HDL cholesterol were also noted. This work opens up the field to new experimental work and represents a clear advance in the field.

We thank you for the comprehensive review and positive evaluations of our manuscript and for providing us with valuable suggestions.

The weaknesses in this study that you have pointed out also point other reviewers have raised, and we addressed them in the revised manuscript.

To remedy this weakness, we evaluated the expression levels of the candidate genes discovered in our study using gene expression data from spinal ligament tissue and related tissues/cells. We used available deposited data in a recently published paper (PMID: 35984875): Gene expression data in a spinal ligament in patients with OPLL and cervical spondylotic myelopathy (CSM) (GSE126060) and in human ligament cells following chondrogenic differentiation (GSE126060). Using these data, we examined the expression levels of the candidate genes of interest in our study. As a result, we confirmed that most genes are expressed in the spinal ligament of OPLL patients and chondrogenic differentiated cells. Furthermore, we confirmed that *WWP2*, the gene of particular interest in our study, is significantly upregulated in the spinal ligament of OPLL patients compared to CSM patients and that *WWP2* is also significantly upregulated in chondrogenic differentiated cells compared to the control group. We have added these to the results and Methods section. Aside from gene expression analyses, we recruited another data set and confirmed genetic associations in the original submission.

To further reinforce the weaknesses of our study, we evaluated candidate gene expression using single-cell RNA sequencing data from available mouse Achilles tendon ossification models (GSE126060 and GSE188758). As a result, we could confirm gene expression at single-cell resolution.

We believe that by improving on this important point that you have pointed out, the findings of this study are now more robust. We would like to thank you again for your valuable review comments.

The main points are covered in the public review. In addition, please consider the following:1. BMI is a fairly sloppy phenotype. It does not really reflect adiposity (eg some highly trained athletes have high BMIs but very low body fat %). Did you either adjust for stature or use another measure of obesity, and if so, what were the results?

Thank you very much for the important comments. We understand that we need to evaluate not only BMI but also other obesity traits. Therefore, we added MR using data from the Giant Consortium and UKBB meta-analysis on the waist-to-hip ratio (WHR), which is believed to reflect adiposity more than BMI. As a result, we confirmed a causal relationship between WHR and OPLL.

p17, L399

“Next, we conducted additional MR for obesity-related traits. We used the data from the large GWAS meta-analyses for obesity-related traits from UK Biobank (UKBB) and Giant Consortium: BMI, waist-to-hip ratio (WHR), and WHR adjusted by BMI (WHRadjBMI) (Pulit et al., 2019) (Supplementary Table 20). Regarding causality from BMI to OPLL, all four MR methods showed significant causality for ALL- and T-OPLL, while only IVW method showed causality for C-OPLL (Supplementary Figure 25, Supplementary Table 21). Regarding causality from WHR to OPLL, 2/4 and 3/4 MR methods showed significant causality for ALL- and T-OPLL, respectively, but no significant results were obtained for COPLL by either method. For both traits, the magnitude of the causal effect size estimated by MR tended to be larger for C-, ALL-, and T-OPLL, in that order, suggesting a strong influence of obesity on the development of T-OPLL. On the other hand, no significant causal relationship was found in the MR on OPLL from WHRadjBMI, a surrogate index of abdominal adiposity. It suggests that systemic obesity, rather than a simple high percentage of abdominal fat, influences the development of OPLL.”

In addition, large GWASs using UKB data (GWAS round 2, http://www.nealelab.is/uk-biobank) for other obesity-related traits existed, so we conducted MR as well. However, MR for these traits could not be estimated accurately because the results were considered highly biased based on the MR-Egger intercept, as shown in Author response table 1. Therefore, causal estimation for these traits could not be done, and we did not add them to the paper.

**Author response table 1. sa2table1:** Estimates of Egger intercept.

Trait	OPLL type	Egger_intercept (se)	p
Body fat percentage	ALL	-0.0285 (0.0107)	7.82E-03
Body fat percentage	C	-0.0120 (0.0148)	4.16E-01
Body fat percentage	T	-0.0672 (0.0160)	3.48E-05
Leg fat percentage (right)	ALL	-0.0302 (0.0084)	3.98E-04
Leg fat percentage (right)	C	-0.0163 (0.0118)	1.71E-01
Leg fat percentage (right)	T	-0.0608 (0.0125)	2.10E-06
Leg fat percentage (left)	ALL	-0.0304 (0.0084)	3.51E-04
Leg fat percentage (left)	C	-0.0161 (0.0118)	1.73E-01
Leg fat percentage (left)	T	-0.0606 (0.0125)	2.08E-06
Arm fat percentage (right)	ALL	-0.0272 (0.0092)	3.32E-03
Arm fat percentage (right)	C	-0.0105 (0.0128)	4.12E-01
Arm fat percentage (right)	T	-0.0604 (0.0136)	1.34E-05
Arm fat percentage (left)	ALL	-0.0280 (0.0091)	2.33E-03
Arm fat percentage (left)	C	-0.0121 (0.0127)	3.41E-01
Arm fat percentage (left)	T	-0.0587 (0.0136)	2.15E-05
Trunk fat percentage	ALL	-0.0169 (0.0111)	1.29E-01
Trunk fat percentage	C	-0.0050 (0.0152)	7.46E-01
Trunk fat percentage	T	-0.0478 (0.0168)	4.77E-03

OPLL, ossification of the posterior longitudinal ligament of the spine; ALL, cervical + thoracic + others; C, cervical; T, thoracic.

2. It is notable that some of the BMD associations you observed were with pediatric rather than adult phenotypes. Do you think that this might reflect dysregulated prolongation of processes that normally are temporally constrained?

Thank you very much for the important comments. Please excuse us for making you confused. We do not believe that our results reflect dysregulated prolongation of processes that normally are temporally constrained. This is because OPLL is a disease that occurs predominantly in the 50s and beyond, which is a time apart from childhood.

Bone density is low at birth, increases gradually with age, peaks in the 20s, and then slowly declines (PMID: 27766484). Therefore, it is assumed that bone density in children reflects the degree of bone formation, although it is affected by various factors. Consequently, we believe that the results of this analysis, which relate to bone density in childhood, are data that support that OPLL is associated with bone formation. This point has been made clear in the Discussion section as follows.

p21, L485

“A gene set enrichment analysis identified significant enrichment in sets associated with BMD in children as well as adults. BMD is low at birth, increases gradually with age, peaks in the 20s, and then slowly declines (Iglesias-Linares et al., 2016). Therefore, it is assumed that BMD in children reflects the degree of bone formation, although it is affected by various factors. Our results can support that OPLL is closely associated with bone formation.”

3. Did the analysis of genetic correlation and other conditions include any of the following: serum Ca, serum phosphate, aortic stenosis, injury-related ectopic ossification, or renal function? It might be worth listing all the tested conditions in a supplementary table.

Thank you for your comment. We evaluated all the traits analyzed in the recently published GWAS paper from BioBank Japan (PMID: 29403010, 32514122, 28892062, and 32152314). Among them, serum Ca, serum phosphate, and renal function were included and analyzed in the previous submission. We had included a list of results in the supplemental figures, but we think the explanation in the text was not clear. Therefore, we revised the manuscript and flagged at first mention in the method and result of genetic correlation that there was a supplementary table listing the complex traits considered as follow.

p14, L306

“We investigated their relationship with OPLL using the GWAS data. We first calculated the genetic correlation between OPLL and 96 complex traits (mean number of around 130K) (Akiyama et al., 2019, 2017; Ishigaki et al., 2020; Kanai et al., 2018) (see Supplementary Table 13 for the traits analyzed).”

p31, L738

“We estimated the genetic correlations using a bivariate LD score regression (Bulik-Sullivan et al., 2015) using only recently published GWAS results for Japanese: 96 complex traits (Akiyama et al., 2019, 2017; Ishigaki et al., 2020; Kanai et al., 2018) (see Supplementary Table 13 for the traits analyzed).".

Injury-related ectopic ossification and aortic stenosis were not analyzed because they were not included in the above GWASs. We think it is a limitation of this study that only 96 traits out of the many diseases in the world have been evaluated for genetic correlation, so we mentioned it in the paragraph summarizing the limitations in the Discussion section.

p23, L537

“Another limitation is that the Japanese GWAS data used for genetic correlation analysis were limited to 96 traits, and other traits not analyzed for genetic correlation may be associated with OPLL.”

4. I do not understand the distinction between diseases and quantitative traits in figure 3. Were the "diseases" coded categorically while the quantitative traits were classified continuously? You list "osteoporosis" as the disease, but most analyses are done on BMD (quantitative trait) or fracture. I suggest merging the 2 parts of the figure and omitting any attempt to distinguish diseases from traits. I like the way this figure displays effect size and statistical robustness together.

As you pointed out, we reconsidered that it would be better to eliminate the distinction between disease and quantitative traits and integrate the results. We have revised the associated figures (Figure 3), tables, and text.

As a result of this correction for multiple testing, osteoporosis just barely did not meet the significance level (P=0.021, FDR=0.0504), but we included it in the MR evaluation because there have been several previous reports of a strong trend toward whole-body ossification in OPLL patients (30113323, 27903251, 30243519, and 6612370).

Reviewer #3 (Recommendations for the authors):Here the authors have conducted a comprehensive genetic interrogation of the condition OPLL. The authors have shown that this disease is heritable and identified some interesting genome-wide significant loci – although these explain only a small proportion of the overall heritability. Their functional genomics analyses add to the paper, and their results are plausible – in that some of the identified genes are involved in bone metabolism (although they also show a relationship with immune/haematopoietic cell grouping, which is not explored or discussed). Further, that there is support for a genetic relationship between BMI and OPLL in the Japanese population which corresponds to the clinical epidemiology.Through no fault of their own, some of the authors' ability to perform and extrapolate gene expression analyses is hampered by a lack of transcriptomic datasets from arguably the most relevant tissue types (including bone, cartilage, ligamentous tissue) within the available datasets worldwide. Similarly, although the authors aimed in their MR analysis to use Japanese GWAS data, they were often constrained to needing to use other ethnic groups (e.g., for the BMD analysis) and thus the MR analyses are not uniform.The authors have also restricted some of their analyses to 'the usual suspects' (e.g., BMI, T2DM, BMD) and not considered other ossification processes such as ankylosing spondylitis and heterotopic ossification. This seems to be unnecessarily limiting and counterproductive when considering the benefits of hypothesis-free approaches in generating new insights into aetiopathogenesis and future therapeutic directions.The discussion needs more nuance and finesse, including placing results in context and considering strengths and weaknesses more carefully. Additionally, the authors have focused on new loci throughout this manuscript: in considering overall pathogenesis/aetiology the authors should include known loci in the discussion. The authors are encouraged to think about their results outside a bone filter.

Thank you very much for the detailed review of our paper and for giving us very valuable comments.

We agree that the lack of transcriptome datasets from the most relevant tissue types (bone, cartilage, ligament tissue, etc.) is a limitation of this study. Thus, in the revised manuscript, we have sought publicly available transcriptome data of OPLL-related tissues and found reasonable expression of the genes we found in relevant tissues. We have added these results in the Result section and revised the Discussion section to state the lack of experimental evidence to consolidate the associations as a limitation of this study. We also added a limitation in the Discussion section: we had to use data from other ancestries in MR.

Regarding the selection of phenotypes to be tested in MR, we agree to select phenotypes in an unbiased manner. Please note that this study used data from all traits examined in a representative BioBank Japan, the largest Japanese biobank, reported in the past, and calculated genetic correlation (GC). Then, based on the results, we have limited the number of traits for which MR is performed in a hypothesis-free manner. Therefore, we do not believe that there is any bias here, but since the traits examined in the Japanese GWAS are limited to about 100 traits out of many, other traits not analyzed for GC in this study may actually be associated with OPLL. This is also a limitation of this study and has been added to the text. In addition, there were no appropriate data for heterotopic ossification, but a large GWAS had been reported for ankylosing spondylitis in the past, so we added an analysis using that data.

We have discussed the known loci as you have suggested regarding the Discussion section. In addition, we also discussed new findings outside of bone metabolism.

Summary to Authors1. There are some methodological issues that the authors cannot solve (e.g., lack of transcriptomic sets from all relevant tissue types; lack of GWAS in Japanese populations for some of the MR analyses); these are iterated in the methods and results but should be highlighted as part of the limitations in the discussion.

Thank you for your very important remarks. As you pointed out, there are several limitations to this study. We have summarized those limitations in one paragraph in the Discussion section.

p22, L525

“There are some limitations in this study. The first is the GWAS sample size. Although this study is the largest OPLL GWAS in the world and used samples collected from facilities throughout Japan, additional samples are needed to clarify the pathogenesis of OPLL further and better define subtype differences. In addition, OPLL GWAS in other ethnic groups has not been reported. We expect future international meta-analysis using data from non-Japanese ethnic groups to elucidate regional differences in the frequencies of OPLL. Second, gene expression data in spinal ligament tissue is scarce. Therefore, we were forced to use tissue data different from ligament tissue, such as GETx (GTEx Consortium, 2015), which did not include bone, cartilage, or ligamentous tissue, and data from a mouse Achilles tendon ossification model, which resembles spinal ligament ossification. We also used bulk data from spinal ligaments, but the sample sizes were limited. It is desirable to increase the number of spinal ligament tissue samples and perform expression analysis using scRNAseq in the future. If so, genes other than those focused on in this study could be identified as causal genes. Another limitation is that the Japanese GWAS data used for genetic correlation analysis were limited to 96 traits, and other traits not analyzed for genetic correlation may be associated with OPLL. In addition, some of the data used for MR were not East Asian. While we cannot immediately improve upon the limitations listed above, we intend to strengthen these areas for future research for OPLL.”

2. Most results appear robust (apart from T-OPLL data). Some additional analyses (specifically, some further co-heritability analyses) are suggested that would strengthen this manuscript and should be easily do-able within their dataset. The discussion needs some redrafting according to the comments below.

As you indicated, we have revised the text to tone it down for interpretation of the T-OPLL results as follows.

p12, L266

“Of these loci, one in the C-OPLL analysis and nine in the T-OPLL analysis were not identified in the analysis of ALL-OPLL and other OPLL subtypes. However, most of the lead SNPs in these significant loci were rare variants. We cannot determine that these are the causal variants based on the present results alone, but there was an interesting variant among them. rs74707424, a leading SNP in the significant locus (19p12), is located in the 3’untranslated region of the ZBTB40 gene. In a recent study using primary osteoblasts of mouse calvaria, Doolittle et al. reported that Zbtb40 functions as a regulator of osteoblast activity and bone mass, and knockdown of Zbtb40, but not Wnt4, in osteoblasts drastically reduced mineralization (Doolittle et al., 2020).”

Furthermore, we conducted additional analyses: recruiting an additional data set to confirm the genetic associations and taking hip-waist-ratio as a phenotype for MR to show the causal relationship between adiposity and OPLL, and using transcriptome analysis to show expression of the genes we found in OPLL-relevant tissues. The discussion is redrafted according to your comments below.

Introduction and methods1. Why is the epidemiology of OPLL in Japanese individuals so different from other ethnicities? A three-fold difference in a polygenic disease between ethnicities is a massive difference. Although there may not be an answer here, this massive epidemiological difference needs more emphasis.

Thank you for the comments. The reason for the difference in incidence among ethnic groups is unknown at this time (and not the topic to address in this paper, as you suggested). As you suggested, the massive epidemiological difference is more emphasized in the introduction.

p7, L135

“OPLL is a common disease; however, its frequency varies depending on the region of the world; high in Asian countries (0.4-3.0%), especially Japan (1.9-4.3%), compared with Europe and the United States (0.1-1.7%) (Matsunaga and Sakou, 2012; Ohtsuka et al., 1987; Sohn and Chung, 2013; Yoshimura et al., 2014).”

2. Are there any previous papers assessing heritability for OPLL? Some data regarding genetic epidemiology would add to this paper. Is there any evidence, for example, of a major gene effect on a polygenic background that might explain the ethnicity differences?

Thank you for the comments. There are no previous papers evaluating the heritability of OPLL such as twin studies. We tried to identify the possible explanation for ethnic differences in the prevalence of OPLL using polygenic signals with our data, but we could not obtain any evidence of a major genetic effect of polygenic background that would explain ethnic differences. We added these points to the Introduction and Discussion sections.

p7, L153

“Furthermore, because of the high incidence within families and close relatives in previous epidemiological studies, genetic factors have long been considered in OPLL development (Matsunaga et al., 1999; Sakou et al., 1991; Terayama, 1989), although there are no previous papers evaluating the heritability of OPLL such as twin studies.”

p22, L508

“This study demonstrates that OPLL is a highly heritable disease. Although previous clinical studies have suggested that OPLL is heritable, there have been no studies that have mentioned the heritability of OPLL. This study is the first to clarify the high heritability of OPLL quantitatively. Further elucidation of the pathogenesis of OPLL from a genetic approach is expected.”

p22, L525

“There are some limitations in this study. The first is the GWAS sample size. Although this study is the largest OPLL GWAS in the world and used samples collected from facilities throughout Japan, additional samples are needed to clarify the pathogenesis of OPLL further and better define subtype differences. In addition, OPLL GWAS in other ethnic groups has not been reported. We expect future international meta-analysis using data from non-Japanese ethnic groups to elucidate regional differences in the frequencies of OPLL.”

3. Three small GWAS, all conducted in a Japanese population, have been meta-analysed. What is the power of each GWAS individually and for meta-analysis overall? There is a large and differential dropout at the QC level (controls >> cases – e.g. set 1 – 92% of cases passed, 85% of controls; set 3: 97% of cases vs. 86% of controls) – Supplementary figure 1. The dropout is unusually high and the reason for such differing rates is unclear, although the reviewer notes (lines 386-396 and Supplementary Table 1) that cases and controls were genotyped separately and in different laboratories (here the reviewer notes the high number of principal components used in the analysis – to ten). The authors should include this issue in their limitations. What where the co-variates used in the GWAS (?BMI ?weight ?presence/absence of T2DM) should be detailed. An Rsq>0.3 for imputation is generous; a more stringent threshold would be 0.8. could the authors justify this choice?

Thank you for the comments.

Regarding a power analysis, as you suggested, we performed a power analysis for ALL-OPLL using GWAS results for each of the three data sets and meta-analysis results.

We examined the statistical power for MAF and odds ratio of variants to show a significance level of P-value = 5E-8 varying allele frequencies and odds ratio. As a result, using the results of the meta-analysis, all had a statistical power greater than 0.5. On the other hand, the results for each data set alone showed low statistical power (Author response image 1 and Figure 1—figure supplement 4). Since meta-analysis increases the statistical power by integrating data that would have low power on their own, we believe this result is reasonable. At the same time, we considered the results of the power analysis for each dataset alone to be meaningless and only included the results of the meta-analysis in the revised manuscript.

**Author response image 1. sa2fig1:** Statistical power analysis. X- and Y-axes represent minor allele frequencies and ORs, respectively. Α-error rate and statistical power are set to 5e-8 and 0.8 (red line) or 0.5 (blue line), respectively. Dots represent ORs of 14 OPLL-associated variants in GWAS meta-analysis for ALL-OPLL. We conducted analyses on each dataset and on the meta-analysis data: (a) Set 1, (b) Set 2, (c) Set 3.

Regarding QC, we noticed that the original submission contained some errors in numbers before QC. We sincerely thank the reviewers for pointing this out. We corrected these errors in the revised manuscript. As a result, the drop-out ratio is now as follows: 7.6 % of cases and 0.3 % of controls in set1; 9.6 % of cases and 0.3 % of controls in set 2; 2.9 % of cases and 5.2 % of controls in set 3 (Supplementary Figure 1). We modified them in the revision, and the differences became much smaller. While we still observed slight differences in passing rates of QC between cases and controls, it seems reasonable because control samples which previously used for other studies (in other words, we used a control data set previously filtered in prior) were subjected as controls in the current study and the majority of case drop-out is due to kinship (duplicated samples or familial cases with other cases – these were checked for controls in prior).

Regarding covariates, we used the top 10 PCs as covariates in our GWAS. This was mentioned in the previous submission. We have revised it and included it in the Methods section for better clarity.

While there is no golden standard for a threshold of imputation scores for meta-analyses, we believe that Rsq > 0.3 is a common threshold, in accordance with our team's previous reports (PMID: 34116867, 31417091, 33272962, 34850884, etc.). Importantly, all of the significant regions in ALL-, C- , and T-OPLL showed Rsq more than 0.638, 0.727, and 0.376, respectively.

This supports the validity of our findings, especially in ALL-OPLL.

4. Are there any other GWAS in other ethnicities? [perhaps not, given how specific this disease appears to be to Japan, and if this is the case it should be mentioned in the discussion. However, if so, this would make the meta-analysis more informative].

Thank you for your question. There are no reports of OPLL GWAS in other ethnicities. Since discovering the causes of ethnic differences is a future task, we have added the following statement to the Discussion section.

p22, L528

“In addition, OPLL GWAS in other ethnic groups has not been reported. We expect future international meta-analysis using data from non-Japanese ethnic groups to elucidate regional differences in the frequencies of OPLL.”

5. Figure 1: Manhattan plot for the meta-analysis. Here the authors appear to have somewhat selectively chosen to annotate some of their significant loci and not others. Line 448 suggests that this is because these are the novel loci. The reviewer thinks it would be more appropriate for all genome-wide significant peaks to be similarly annotated (perhaps in different colours for new vs. known loci), to give the reader the full view of the genetic information within this one image rather than having to refer to previous publications.

Thank you for your suggestion. As you suggested, all genome-wide significant peaks are annotated (Figure 1).

6. The reviewer notes that the criteria for cases vary between sets 1 and 2 vs. set 3. The authors do not really justify their decision to split the data into cervical and thoracic subsets from an epidemiological (genetic or otherwise) or clinical basis. Are these really discrete clinical entities, with separate epidemiology and disease course? The retro-assessment of C-OPLL and T-OPLL is not detailed (one reviewer? More than one reviewer? Quality control for calling this clinical phenotype?).

Thank you for the comments. We agree with the necessity to justify the separation of cases for COPLL and T-OPLL. In recent years, many clinical studies have been published showing that the clinical characteristics of OPLL differ between the cervical and thoracic types (PMID: 31290005, 35245901). This content is added to the text to emphasize the significance of analysis for each subtype.

In addition, it has been reported that T-OPLL in particular has a strong tendency to ossify and is often severe (PMID: 31290005, 35245901). So then, in recent years, we have focused on T-OPLL among OPLL cases and collected genomes. Therefore, cases in the set 3, which were collected recently, are only T-OPLL. This is why the criteria for cases vary between sets 1 and 2 vs. set 3.

p7, L140

In recent years, OPLL has been reported to have different clinical characteristics depending on the affected region: higher body mass index (BMI), earlier-onset of symptoms, and more diffuse progression of OPLL over the entire spine in the thoracic type of OPLL (T-OPLL) than in the cervical type (C-OPLL) (Endo et al., 2020; Hisada et al., 2022).

The evaluations of the OPLL images were performed at each facility, and it is impossible to trace how many orthopedic surgeons performed the evaluations at each. However, we believe the data are reliable because spine specialists among orthopedic surgeons performed the evaluation.

7. Supplementary Figure 3 – the locus zoom plots do not show which SNPs are imputed and which are genotyped. This should be added to the diagrams.

Following your suggestion, we have modified the locus zoom plots to distinguish between genotyped SNPs and imputed ones. (Supplementary Figure 3-c).

8. Line 461 Decisions around the associated locus of a lead variant is usually set by LD rather than physical distance.

Thank you for the comments. We believe that defining a region by distance rather than LD is a common method since the recent increase in sample size resulted in many significant signals (PMID: 34116867, 31417091, 33272962, 34850884, etc.). However, as you point out, it is important to assess by the LD calculations that each significant locus is independent. We defined regions by physical distance and then confirmed that there are no SNPs outside of each locus in LD (r2 > 0.1) with SNPs that met genome-wide significance levels within the locus.

p26, L620

“We confirmed that there are no SNPs outside each locus in LD (r2 > 0.1) with SNPs that met genome-wide significance levels within the locus.”

9. Line 488 – Gene set enrichment analysis – although this is a reasonable process, either here or elsewhere the authors should provide reference justification for choosing to assess genes based on physical location (within 1Mb of leading SNP plus up to five genes 'closest' to leading SNPs) as there are now notable examples in the literature where the leading SNP signal does not correlate physically to the causative gene, with obesity being a lead example here.

Thank you for your comment. While there are cases that distant genes are responsible for associations (as the reviewer suggested), the closest genes are still one of the best candidates of responsible genes for associations (even in the ABC paper, the closest genes show very good predictability of responsible genes, PMID: 33828297, 31784727). We added references to justify the selection of genes. In addition, we added to the Discussion section possible other responsible genes different from the closest genes.

p28, L665

“Because variants often act on genes that are close in the distance (Fulco et al., 2019; Nasser et al., 2021), we selected genes on a distance basis using the following criteria and used them as input data: genes (i) located within 1 Mb and (ii) the five closest to the leading SNPs of each genome-wide significant locus.”

10. Line 498 – eQTL analysis in 'all available tissues' – the authors need to comment in their discussion on the relevance/appropriateness of the tissues captured in GTEx v8 (noting here bone and/or cartilage and/or ligamentous tissue are not included in GTEx v8).

Thank you for the comments. We agree to mention tissues included in the GTEx. Since this is one of the limitations of this study, we have included the following in the limitation part of the discussion.

p22, L530

“Second, gene expression data in spinal ligament tissue is scarce. Therefore, we were forced to use tissue data different from ligament tissue, such as GETx (GTEx Consortium, 2015), which did not include bone, cartilage, or ligamentous tissue, and data from a mouse Achilles tendon ossification model, which resembles spinal ligament ossification.”

11. Line 503 – unclear why v7 is used here whereas v8 is used above.

We used GRCh37 as our reference genome sequence throughout this study. Therefore, in SMR, we downloaded and used data from GTEx v7, the same build as in this study, instead of v8 (GRCh38). We have added an explanation of the above in the Method section.

12. Line 518ff Genetic correlations were assessed using bivariate LD score regression and Japanese data – it is implied here that this only used Japanese data (which is appropriate) but this should be definitively confirmed.

As you suggested, we modified the manuscript to clearly state that we used only Japanese data.

p31, L738

“We estimated the genetic correlations using a bivariate LD score regression (Bulik-Sullivan et al., 2015) using recently published GWAS results for Japanese: 96 complex traits (Akiyama et al., 2019, 2017; Ishigaki et al., 2020; Kanai et al., 2018).”

13. Lines 528ff for the MR study, the authors used GWAS data from varying ethnic backgrounds, including Japanese and European and east-Asian (presumably Chinese+Japanese here), which varied from trait to trait (e.g European GWAS was used for the BMD analysis, and east Asian for the T2DM analysis). This limits the robustness of this section, as these were not equally valid tools for the MR analysis. In particular, the BMI data were informed from both Japanese and European GWAS data. As the relationship between BMI and OPLL is one of the major findings of this paper, this analysis should be repeated using data ONLY from Japanese BMI GWAS, as a sensitivity analysis.

Thank you for the comments. As suggested, we repeated the analysis using data from Japanese BMI GWAS as a sensitivity analysis. As a result, we could confirm the direction of the effect, although the results were not significant for ALL-OPLL. However, the results were significant for T-OPLL, reinforcing the causality of BMI on T-OPLL (Supplementary table 19).

p16, L356

“Based on the above MR results, we focused on the causal relationship between high BMI, that is obesity, and OPLL. First, we repeated MR using only Japanese BMI data (Akiyama et al., 2017) to estimate causal effects on OPLL as a sensitivity analysis. As a result, we confirmed the positive direction of the effect. Furthermore, the results were significant for TOPLL, reinforcing the causality of BMI on T-OPLL (Supplementary Table 19).”

14. Line 555ff – the authors appropriately assessed their data to ensure no overlap between exposure and outcome datasets – but it is not clear what they then did, having found that there were small overlaps.

The message here is that only the control sample of binary (OPLL) phenotype contains duplicated samples, so the data used for our MR were fine. However, the degree of overlap in the sample was emphasized and did not convey what we originally intended to convey. We have rewritten the method to ensure that our message gets across.

p32, L777

“We checked the cohort data used in our MR and found no overlap with the OPLL case sample, although the control samples used in the OPLL study overlapped with up to 2.2% of samples used in the BMI study and 3.4% in the T2D study.”

15. Considering the PRS of BMI wrt application to OPLL GWAS – the authors appropriately use a Japanese-population GWAS here.

Thank you for your feedback to the PRS of BMI.

16. Line 577 – the reviewer notes in line 582 that having tried out differing r2 and p values they eventually chose r2 of 0.9 tho' P-value=0.6 – this would mean almost any result in the BMI GWAS could be included. From the image Supplementary Figure 20 they could have chosen p=0.05 without much difference in their rho value, and this would seem at least a little more stringent.

Thank you for the comments. Sorry for making you confused.

The horizontal line in the figure represents Spearman’s rho between the BMI and the BMI polygenic risk score in validation data. And there is little change in the value of rho for any p-value selected above 0.05. So, as you pointed out, if we choose p-value = 0.05 and do the PRS scoring for test data, I think it would still be a good predictor. However, to perform a rigorous PRS, we believe it is optimal to select r2 = 0.9 and P-value = 0.6, which had the highest correlation (while the differences are small). Therefore, while we appreciate your feedback, we have not made any changes to the PRS methodology.

Results1. Line 175FF: have the authors corrected their p-value for significance by this (modest) genomic inflation factor (1.11)?

Thank you for the comments. As you pointed out, the genomic inflation factor indicated mild inflation of statics in the GWAS (λGC = 1.11). Therefore, to assess whether this elevation was due to polygenicity or biases, we conducted LDSC and checked the intercept. The intercept was 1.03, and we found that the inflation was mostly due to polygenicity with little bias in the results. For this reason, we did not correct our results. This was described in the Results as follows.

p8, L177

“The genomic inflation factor (λGC) was 1.11 and showed slight inflation in GWAS; however, the intercept in linkage disequilibrium (LD) score regression (Bulik-Sullivan et al., 2015) was 1.03, indicating that inflation of the statistics was mainly from polygenicity and minimal biases of the association results (Supplementary Figure 2).”

2. Line 209ff: the authors should make an explicit comment on their lack of enrichment in DXA-assessed BMD in adults (e.g. using data from GEFOS2).

As you suggested, we made the explicit comment on the lack of enrichment of association signals in DXA-assessed BMD in adults.

p10, L219

“However, we observed no significant enrichment in BMD in adults measured by dual-energy X-ray absorptiometry in this analysis.”

3. Line 219ff: the authors' decision as to which are and which are not plausible causative genes seems arbitrary [based mainly on previous data for monogenic skeletal dysplasias it would seem]. If only these genes for which there is prior knowledge are considered it somewhat defeats the point of trying to gain new knowledge from hypothesis-free approaches. This continues to be an issue in the discussion too. The authors should describe the known function of other genes too.

Thank you for your very important remarks. We are afraid we confused you about plausible causative genes in the original submission.

We should interpret the associations after detecting association signals. To prioritize genes potentially responsible for associations, there are various ways shown in the previous studies, including gene functions, distance to lead variants, 3D interaction, and so on.

What we did is to prioritize genes as candidates of causal genes based on various information including gene functions previously known. We did not exclude possibility of other genes as candidates of causal genes. In addition, we cannot prioritize genes based on previous knowledge in the regions where there are no surrounding genes related to bone metabolism.

We made these points clear in the revision. We additionally mentioned in the Discussion section the possibility that genes with no reported function related to bone metabolism may be causal genes.

Additionally, there was an error in the previously reported analysis, and the results have changed slightly, and finally, we discovered three candidate genes: *EIF3E*, *EMC2*, and *TMEM135*. As you indicated, we have made descriptions for all genes.

p10, L225

*“EIF3E* (Eukaryotic translation initiation factor 3 subunit E) encodes a protein that is a component of the eukaryotic translation initiation factor 3 (eIF-3) complex, which functions in and is essential for several steps in the initiation of protein synthesis (Lee et al., 2015; Masutani et al., 2007). A Proteomics study in a rat model of heterotopic ossification reported that Eif3e was upregulated in ossified tissues and may be involved in tissue ossification by regulating hypoxia-inducible factor (HIF) signaling, which has an important role in osteogenesis (Wei et al., 2022). *EMC2* (endoplasmic reticulum membrane protein complex subunit 2) encodes a part of the endoplasmic reticulum membrane protein complex (EMC) that functions in the energy-independent insertion of newly synthesized membrane proteins into the endoplasmic reticulum membrane, an essential cellular process (Chitwood et al., 2018; O’Donnell et al., 2020). However, basic experiments evaluating the effects of *EMC2* on ligament and bone tissue have not been reported, and the mechanisms involved in OPLL are unknown. On the other hand, this analysis reinforced the possible involvement of *TMEM135* in the development of OPLL.”

p18, L429

“Although the effects of *EMC2* and *RP11-967K21.1* on ligament and bone tissue have not been reported and are unknown, future studies may elucidate the mechanisms and reveal that they are the actual causal genes.”

4. Similarly Line 241ff. Whilst reporting their significant enrichment in the connective/bone cell group, the authors have not mentioned here that they also had significant enrichment for the immune/hematopoietic cell group, with a significant result for skeletal muscle which did not survive after FDR adjustment. These other cell groups are highly relevant to the phenotype and should be mentioned here similarly.

Thank you for your very important remarks.

As you suggested, we additionally mentioned the significant enrichment for the immune/hematopoietic cell group. As for the skeletal muscle cell group, we did not discuss it here because it was suggestive but not significant (P=0.059).

p11, L254

“We conducted partitioning heritability enrichment analyses to investigate cell groups related to OPLL. We observed significant enrichment in the active enhancers of the connective/bone cell group and the immune/hematopoietic cell group (Supplementary Table 9). We then analyzed each cell type belonging to these groups and found significant enrichment of H3K27ac in chondrogenic differentiation cells (Supplementary Table 10). These results concord with previous findings that in OPLL chondrocyte differentiation in the endochondral ossification process occurs (Sugita et al., 2013) and provide new insights into the involvement of immune system cells in OPLL development, which has received little attention to date.”

5. The authors' choice for co-heritability studies is limited. The authors have not considered GWAS for AS, for example, which may be highly relevant. The authors should consider an unbiased coheritability analysis rather than only the 'usual suspects' – as the latter will only really confirm prior thoughts and not illustrate novel pathways/mechanisms (for example see the recent paper on IBS).

Thank you for your comment. As noted above, since we analyzed all the representative Japanese traits in an unbiased manner for which GWAS was performed, we believe that there is little bias.

On the other hand, the fact that we analyzed only some of the numerous traits that exist in the world is a limitation of this study, and we have added this point in the Discussion section. In fact, there are no GWAS of AS in Japanese.

However, as you pointed out, adding an AS analysis would be nice. We took the data from Ellinghaus *et al.* 2016 and compared the SNP effect sizes with those in our OPLL GWAS, assuming fixed effects across populations. Please note that we could not perform GC analysis using LDSC due to ethnical differences. As a result, no significant results were observed (Figure 1—figure supplement 17).

6. Supp Figure 5: the calling of eight new loci for T-OPLL is over-enthusiastic. There is very little genetic support for ZBTB40, for example (see Locuszoom plot in Supplementary Figure 8), and it's not clear whether this is a genotyped or imputed SNP (not shown in Supplementary table 11). Similar comments are made about the alleged novel loci in plots c, e, f, g, and h. The only believable putative novel loci are shown in plots b and d. The authors should tone down their claims about discrete novel loci for T-OPLL substantially, therefore, throughout the manuscript (including abstract, results, and discussion). Related to this, the authors should reconsider rewriting sections where they feature T-OPLL. In discussing these possible site-specific T-OPLL GWAS results, the authors do not avail themselves of the opportunity to discuss the two plausible loci but only discuss ZBTB40. What is known about genes within the two more plausible loci?

We think this is the same point as the one you raised earlier. As you suggested, the claims about discrete novel loci for T-OPLL are substantially toned down throughout the manuscript. The sections where we featured T-OPLL are rewritten.

7. Line 265ff the authors should flag at first mention here and in the methods (at line 520) that there is a supplementary table listing the complex traits considered here, especially as this is limited (appropriately) to previous genetic studies in Japanese populations (as per the methods). Although the authors' approach here is appropriate (i.e., limited to the Japanese population), the point is that these GWAS represent only a subset of the many traits mapped by GWAS internationally [this is given later in the paragraph but should be flagged earlier for the reader].

Thank you very much for the important comments. As you suggested, we flagged at first mention here and in the methods that there is a supplementary table listing the complex traits considered.

p14, L306

“We investigated their relationship with OPLL using the GWAS data. We first calculated the genetic correlation between OPLL and 96 complex traits (mean number of around 130K) (Akiyama et al., 2019, 2017; Ishigaki et al., 2020; Kanai et al., 2018) (see Supplementary Table 13 for the traits analyzed).”

p31, L738

“We estimated the genetic correlations using a bivariate LD score regression (Bulik-Sullivan et al., 2015) using only recently published GWAS results for Japanese: 96 complex traits (Akiyama et al., 2019, 2017; Ishigaki et al., 2020; Kanai et al., 2018) (see Supplementary Table 13 for the traits analyzed).”

8. Line 271: ossification and bone mineral density accrual are quite different processes. In particular, BMD is likely to be artefactually elevated by the presence of OPLL, if this affects lower spinal areas, as it is also in AS, DISH, etc. The authors need to provide more clarity, nuance, and detail here.

Thank you very much for the important comments. We agree to add a description for the relationship between ossification and BMD and for the potential artifact in the measurement of BMD. We have provided a more detailed explanation of why we focus on the link between BMD and OPLL.

p14, L316

“The result of genetic correlation analysis for osteoporosis barely did not reach the false discovery rate (FDR)-corrected significance level, but we included it in the MR evaluation because there have been several previous reports of a strong trend toward whole-body ossification in OPLL patients (Hukuda et al., 1983; Mori et al., 2016; Nishimura et al., 2018; Yoshii et al., 2019). In the analysis, we used BMD, the main diagnostic criteria item for osteoporosis. BMD in the spine may reflect artifacts from OPLL itself, but higher BMD was also reported in patients with OPLL in the femur and the radius, a non-weight-bearing bone (Sohn and Chung, 2013; Yamauchi et al., 1999).”

9. The results would benefit from editorial comments being removed from the results and placed in the discussion (e.g. lines 269-271).

This content was already mentioned in the discussion and was removed because it would have been redundant. This part of the description has been removed.

10. Line 316ff PRS scores – these β values may be significant but they are small. Can the authors contextualize their results with, say, the PRS for BMI and T2D?

Thank you very much for your comment. As you suggested, we have compared the effect size of BMI PRS on T2D, for which BMI is known to be one major cause, with that on OPLL.

We generated PRS as the sum of risk alleles weighted by the log odds ratio. However, at the time of the last submission, the sum (PRS) was not scaled and we used the raw data for subsequent analysis. In this way, the magnitude of the β values is not directly interpretable. Then, after standardizing the PRS, we reran our subsequent analysis. The results are the same as before except for the y-axis values (Figure 5). Now the β of 1 SD of BMI is over 0.3, equivalent to OR of 1.35, which is a substantial increase in risk.

Next, we conducted BMI PRS scoring for the T2D cases and controls, resulting in a positive effect size of BMI PRS on T2D (β = 0.099, 95% CI: 0.087- 0.110). This T2D data set overlaps almost completely with the data from the discovery phase of the BMI PRS, then it carries a high risk of statistical inflation due to model overfitting. However, since no other data was available for T2D, the comparison was made in this manner, knowing the above points.

Comparing this result with that of OPLL, we found that the effect of BMI PRS on OPLL was higher than T2D, so we have added the following in our revised manuscript.

p17, L386

“Further, the effect sizes of BMI PRS on OPLL were all larger than that on T2D analyzed in a similar manner (Method) (β = 0.099, 95% CI: 0.087- 0.110), indicating that high BMI is a major cause of OPLL.”

p34, L822

“To compare the effect size of BMI PRS in OPLL with T2D, for which BMI is known to be one of the major causes, we performed the same BMI PRS scoring on the T2D data set (39,758 cases and 111,487 controls). Unfortunately, the T2D data set overlaps almost entirely with the data from the discovery phase of the BMI PRS, which carries a high risk of statistical inflation due to model overfitting. However, since no other data was available for T2D, the comparison was made in this manner, knowing the above points.”

11. The authors have attempted to split their data by clinical phenotype (cervical vs. thoracic OPLL); the data supporting multiple novel loci for T-OPLL do not appear robust from the locuszoom plots and the authors are encouraged to tone down this component substantially (for example, focusing on ZBTB40 as a locus for T-OPLL appears implausible from the Locuszoom plot (one is left with an impression that there was some prior hypothesis contribution to this point [e.g., it being a known BMD locus]) whereas other much more plausible loci (based on the Locuszoom images) do not receive similar attention).

Thank you for your comment. As you pointed out earlier, we have toned down the description for signals found in the OPLL subtype.

DiscussionTheir discussion is inadequate; and needs much more care and non-biased interpretation, and detailing of the [many] limitations. Also, there is an obvious clinical question, that they haven't posed: has weight loss been trialled as a therapeutic approach to mxmt of OPLL?

Thank you very much for your detailed advice. We have taken your advice and modified the Discussion section.

1. The authors' functional assessment in their first paragraph on bone formation needs toning down. Certainly, the authors' results are enriched for signals in bone metabolism pathways – but they don't know that these are necessarily anabolic, or anti-catabolic; and line 324ff bone formation and resorption are normally tightly coupled processes so it would be unusual for these to be uncoupled in OPLL uniquely. Moreover, the authors have mainly featured those genes that suit their argument; and have not commented, for example, that their gene enrichment analyses also identified immune/haematopoietic cell groups. Allowing for the time being their comment to stand: is there any evidence that bisphosphonates (that suppress both bone formation and bone resorption) improve outcomes with OPLL?

As you suggested, we have toned down the functional assessment in the first paragraph on bone formation. Further, a new paragraph was added to the discussion regarding the possible involvement of immune/hematopoietic cells, as discovered by partitioning heritability enrichment analyses. Regarding your last question, one randomized controlled trial evaluated the prognosis of OPLL with a bisphosphonate, but this study did not show significant results (Yonenobu et al., 2006). We added to the manuscript a sentence that further study is needed to determine whether existing drugs such as bisphosphonates are effective or whether new drugs need to be developed.

p22, L513

“Although this study identified that OPLL is closely linked to bone metabolism, there are few studies examining the prognosis of OPLL with the use of bone-modifying agents such as bisphosphonates, which suppress both bone formation and resorption. A randomized controlled trial evaluated the effect of etidronate disodium on postoperative OPLL progression in patients with OPLL who had undergone posterior decompression surgery, but no significant effect was demonstrated (Yonenobu et al., 2006). Future studies are needed to determine whether existing agents are adequate or whether we need to develop new agents.”

2. The relationship between EB and osteoporosis needs far more nuance if the authors are to include it in their argument regarding PLEC: it is thought to be multifactorial and related more to nutrition, drugs, mobility, etc, and there is not, to my knowledge, a known underlying mechanistic link directly related to the mutation and bone per se.

As you describe, bone density has been reported to decrease in patients with EB due to various effects: reduced mobility, compromised nutritional intake by oral and gastrointestinal tract complications, low 25(OH) vitamin D levels caused by reduced ultraviolet rays access to the skin, and increased osteoclastic activity by inflammatory cytokines (PMID: 30118182). On the other hand, the underlying biological mechanism by which *PLEC* mutations affect bone metabolism is not clear. We have added these descriptions.

p18, L426

“The *PLEC* mutation causes epidermolysis bullosa, in which osteoporosis is one of the main comorbidities. Although various causes have been reported to induce osteoporosis in this disease (Chen et al., 2019), the underlying biological mechanism by which *PLEC* mutations affect bone metabolism is unclear, and future studies are expected to elucidate this mechanism.”

3. The benefits of considering this approach as novel knowledge-generating have been somewhat lost by the authors only focusing on bone in their discussion, and as they mostly do not mention information about previously reported loci it's hard to contextualise their newer findings with older knowledge regarding bone pathways vs. other pathways in terms of OPLL overall.

As you suggested, we have added a discussion of the previously reported loci.

p18, L411

“Although many unknowns exist regarding the function of the six regions found in the past, functional analysis of chromosome 8 (q23.1), where *RSPO2* is located, has progressed. rs374810, the most significant variant in the present and the past study, is located on the promoter region of *RSPO2* in chondrocytes. Its risk allele (C allele) causes a loss of transcription factor C/EBPβ binding; therefore, *RSPO2* expression is reduced. Finally, the Wnt-β-catenin signal that blocks chondrocyte differentiation of ligament MSCs is suppressed, which triggers OPLL generation (Nakajima et al., 2016).”

As we responded to the previous comments, we did not restrict candidates of responsible genes to genes related to bone metabolism. What we did is to try to interpret the association signals by various information including gene functions. We did not exclude the possibility of other genes as responsible genes. We made this point clear in the revised manuscript.

p18, L429

“Although the effects of *EMC2* and *RP11-967K21.1* on ligament and bone tissue have not been reported and are unknown, future studies may elucidate the mechanisms and reveal that they are the actual causal genes.”

Also, as noted above, we have added a paragraph discussing the involvement of tissues other than bone.

p19, L435

“Partitioning heritability enrichment analyses by LD score regression identified that OPLL GWAS signals enrich not only the active enhancers of the connective/bone cell group but also those of the immune/hematopoietic cell group. It is well known that heterotopic ossification of ligaments and tendons, in which mesenchymal stem cells differentiate into osteochondrogenic cells rather than myocytes or tenocytes, is triggered by tissue damage due to trauma (Convente et al., 2018; Torossian et al., 2017). Tissue injury triggers a systematic inflammatory response with the mobilization of neutrophils, monocytes, and lymphocytes at various stages of inflammation. Monocytes/macrophages are involved in abnormal wound healing and influence the development of heterotopic ossification (Sorkin et al., 2020). This fact suggests that immune system cells are involved in the tissue repair process of the posterior longitudinal ligament tissue of the spine, which is the host that causes ossification in OPLL as well. In addition, gene expression analysis using scRNAseq data confirmed that the candidate genes discovered in this study were expressed not only in mesenchymal cells but also in immune cells.”

4. Lines 340ff – here the authors should be careful not to conflate BMD with ossification processes, and this section requires more nuance. The relationship between obesity and BMD is extremely contentious, and only referring to Lv et al. is a very simplistic reflection of the literature. (most of this paragraph really discusses the relationship between BMI and BMD, and not, as is pertinent here, BMD and OPLL, so it should be revised to focus on the newer loci presented here).

Thank you very much for your advice. This is very similar to the previous comments by the reviewer. As we responded to the comments above (Reply 5 for the Result section), for heterotopic ossification, there was no GWAS of AS in Japanese. Therefore, we took AS in the European populations and compared the effect sizes between OPLL and AS.

5. The authors have missed an opportunity to consider a relationship between OPLL and other ossifying processes, such as ankylosing spondylitis and heterotopic ossification. Whilst these may not have had GWAS performed in Japanese populations the authors have not always had Japanese population data for those studies that they have included (e.g., BMD).

Thank you very much for your advice. We think this is a point you pointed out earlier (advice 10 for Results). We cannot directly interpret the β coefficients since we did not normalize the PRS. In the revised manuscript, we showed that 1 SD of BMI increased the risk of OPLL with OR of 1.35. We also conducted BMI PRS scoring for the T2D data set and compared this result with that of OPLL, and the effect of BMI PRS on OPLL was higher than that of T2D, indicating that BMI has a substantial effect on OPLL.

6. The PRS suggests that there is a genetic relationship between BMI and OPLL, with BMI driving OPLL. However, the β suggests the clinical impact is extremely small. Contextualising their results with other PRS would help the reader understand the clinical impact of their findings.

Thank you very much for your advice. We think this is a point you pointed out earlier (advice 10 for Results). We cannot directly interpret the β coefficients since we did not normalize the PRS. In the revised manuscript, we showed that 1 SD of BMI increased the risk of OPLL with OR of 1.35. We also conducted BMI PRS scoring for the T2D data set and compared this result with that of OPLL, and the effect of BMI PRS on OPLL was higher than that of T2D, indicating that BMI has a substantial effect on OPLL.

7. The authors do not discuss the strengths and weaknesses of their paper, which would add sobriety to their discussion.

Thank you very much for your advice. We have emphasized only the strengths of our research and have not discussed the weaknesses (limitations) enough. We have added a paragraph to the Discussion section that discusses the limitations of this study.

8. Paradoxically, the authors focus a lot on talking about novel treatments, etc but by only looking at previously known pathways they've limited the ability to look at these. Also, the most obvious way to manage high BMI would be weight loss. Not sure this really represents a novel treatment for OPLL, as such.

As you pointed out, we did not consider enough pathways other than obesity and bone metabolism. In addition to obesity and bone metabolism, we have added to the manuscript a discussion on the involvement of the immune system in OPLL, as noted above.

We believe that the suppressive effect of weight loss on OPLL should be examined in future prospective studies.

p22, L519

“Regarding obesity, the cause of OPLL, weight loss guidance, and aggressive bariatric treatment, including surgery, for some patients may be considered. Long-term prospective studies are needed to evaluate the effect of weight loss on OPLL suppression.”

9. The authors have also not contextualized their results in light of previous data. For example, here they have shown heritability of OPLL of >50%. Have there been any previous studies of heritability?

This study is the first study on the heritability of OPLL. Since demonstrating high heritability is an important finding of this study, we have included the following in the Discussion section. As we responded to the other comments, such as Comment 8 in Discussion, we put additional descriptions in the context of previous results and data in the revised manuscript.

p22, L508

“This study demonstrates that OPLL is a highly heritable disease. Although previous clinical studies have suggested that OPLL is heritable, there have been no studies that have mentioned the heritability of OPLL. This study is the first to clarify the high heritability of OPLL quantitatively. Further elucidation of the pathogenesis of OPLL from a genetic approach is expected.”